# An imputation platform to enhance integration of rice genetic resources

Diane R. Wang [1,6], Francisco J. Agosto-Pérez[1], Dmytro Chebotarov[2], Yuxin Shi[1], Jonathan Marchini [3], Melissa Fitzgerald [4], Kenneth L. McNally [2], Nickolai Alexandrov [2,7] & Susan R. McCouch [1,5]

As sequencing and genotyping technologies evolve, crop genetics researchers accumulate increasing numbers of genomic data sets from various genotyping platforms on different germplasm panels. Imputation is an effective approach to increase marker density of existing data sets toward the goal of integrating resources for downstream applications. While a number of imputation software packages are available, the limitations to utilization for the rice community include high computational demand and lack of a reference panel. To address these challenges, we develop the Rice Imputation Server, a publicly available web application leveraging genetic information from a globally diverse rice reference panel assembled here. This resource allows researchers to benefit from increased marker density without needing to perform imputation on their own machines. We demonstrate improvements that imputed data provide to rice genome-wide association (GWA) results of grain amylose content and show that the major functional nucleotide polymorphism is tagged only in the imputed data set.

[1] Section of Plant Breeding and Genetics, School of Integrated Plant Sciences, Cornell University, Ithaca, NY 14853-1901, USA. [2] International Rice Research Institute, DAPO Box 7777,1301 Metro Manila, Philippines. [3] Department of Statistics, University of Oxford, Oxford OX1 3LB, UK. [4] School of Agriculture and Food Science, University of Queensland, 4072 QLDBrisbane, Australia. [5] Biological Statistics and Computational Biology, Cornell University, Ithaca, NY 14853-1901, USA. [6]Present address: Department of Geography, University at Buffalo, Buffalo, NY 14261, USA. [7]Present address: Inari Agriculture Inc., Cambridge, Cambridge, MA 02139, USA. Correspondence and requests for materials should be addressed to S.R.M. (email: srm4@cornell.edu)

Genetic imputation is an approach to infer genotypes at unobserved sites in a study data set using haplotype information from a typed reference data set. Imputed panels can be used to improve resolution for genome-wide association (GWA) and to integrate data sets[1,2]. Leveraging haplotype diversity information from publicly available cosmopolitan reference panels for imputation has been widely used in human genetics research, beginning with reference panels derived from the first HapMap Projects and then from the 1000 Genomes Project as sequencing technologies improved[3–7]. In contrast to the consortium approach taken by the human genetics community, crop researchers have conventionally addressed resource development in a more distributed fashion. This may be related to the intrinsically flexible nature of plant populations; a single species can exhibit a range of mating habits and environmental adaptations, and individuals can be readily propagated. These features give rise to the many types of plant collections that are genotyped, e.g., biparental and multiparental mapping populations, locally adapted landrace collections, diversity panels, and various generations of elite breeding material. As a result, myriad genomic data sets of varying quality and density on independent collections of samples have accrued. Imputation has the potential to provide a generalized solution to enrich genomic data sets and facilitate data integration across germplasm resources, especially important for sparse data sets generated from platforms such as genotyping-by-sequencing (GBS) or skim sequencing used commonly in plant research[8–10].

Cultivated Asian rice (*Oryza sativa* L.), the first crop genome to be fully sequenced[11,12], has been a pioneer species in genomic research. It is rich in open-access resources, including diverse germplasm samples available as publicly available purified (homozygous) seed stocks[13], data sets from single-nucleotide polymorphism (SNP) array genotyping and resequencing[13–18], and a recent pan-genome initiative that aims to provide a union set of all genes within the species[19]. Despite these features, there currently exists no rice data set that has been vetted to serve as a cosmopolitan reference panel for imputation and no systematic approach to facilitate integration across different resources. One of the specific challenges for the rice community is the fact that *O. sativa* is comprised of deeply differentiated subpopulations across two varietal groups (VG) due to the combined effects of natural and human selection: Indica VG: *indica* (*ind*), *aus* (*aus*); Japonica VG: *temperate japonica* (*tej*), *tropical japonica* (*trj*) and *aromatic* (*aro*). The five subpopulations are distinguishable based on ecological adaptation, grain quality characteristics, and complex physiological parameters, as well as population genetics metrics such as linkage disequilibrium (LD) decay[13].

The most comprehensive *O. sativa* genomic data sets released to date, accompanied by publicly available germplasm, are the 3000 Rice Genomes (3KRG), genotyped using Illumina short-read next-generation resequencing technology[14,20], and the Rice Diversity Panels 1 and 2, ~1500 diverse rice varieties genotyped using a genome-wide high-density rice array (HDRA)[13] (hereafter referred to as the HDRA panel). These panels sample across the geographic and genetic space occupied by *O. sativa*'s five distinct subpopulations. The 3KRG Project yielded a set of 29 M biallelic SNP markers, an 18 M base SNP set, and a 4.8 M filtered, high-quality SNP set as part of its Release 1.0 (http://snp-seek.irri.org/download.zul). The HDRA Panel was assayed using a 700 K SNP array. Collectively, the 3KRG and HDRA Panels are believed to be representative of global *O. sativa* diversity.

Here, we set out to provide easily accessible imputation capacity for rice researchers worldwide. We examine parameter effects on imputation in rice using IMPUTE2 and impute the HDRA Panel to 4.8 M SNPs using 3KRG as the reference panel. We demonstrate the benefits that imputed data can provide to enhance the resolution of rice GWA studies using grain amylose content as a case study. We assemble a Rice Reference Panel (RICE-RP) by merging the HDRA and 3KRG data sets via reciprocal imputation (Table 1, Fig. 1) and develop a web-based service called the Rice Imputation Server to increase imputation accessibility to rice researchers throughout the world. This work paves the way to boosting research efficiencies by enabling investigators within both basic and applied research domains to integrate discrete data sets and to augment marker density, improving the power and resolution of genomic studies in rice.

## Table 1 Description of data sets used in this study

| Data set | Marker number | Sample number | Unique sample number |
|---|---|---|---|
| HDRA | 700,000 | 1568 | 1553 |
| HDRA_filtered | 160,267 | 1568 | 1553 |
| HDRA_imputed | 4,829,392 | 1568 | 1553 |
| 3KRG | 4,817,964 | 3023 | 3023 |
| RICE-RP | 5,231,433 | 4591 | 4481 |

Sample number refers to the number of samples genotyped within each data set. A unique sample number accounts for the fact that biological replicates were included within and across data sets as technical controls. For a graphical overview, see Supplementary Fig. 1

## Results

**Rice imputation with IMPUTE2.** IMPUTE2 was primarily developed for imputation in humans. We carried out a series of experiments to investigate the parameter settings that provide the best performance in rice samples, using a gold standard set of accessions that overlapped between the HDRA and 3KRG Panels as our study panel (individuals, $n = 50$; markers, $m = 160,267$) (Supplementary Fig. 1; Supplementary Data 1). For the reference panel, we used the 3KRG accessions but removed the 50 study panel individuals ($n = 2973$; $m = 4,817,964$) to ensure that there would be no identical samples across reference and study panels. We first assessed whether there was any difference between the use of phased and unphased reference panels. The expectation was that there should be no difference in accuracy due to *O. sativa* being an inbreeding species such that genotypes are generally equivalent to haplotypes. To test this, we targeted three 2-Mb segments across the genome (regions on chr 1: 0.5–2.5 Mb; chr 3: 7–9 Mb; chr 12: 20.5–22.5 Mb) and observed no significant difference between using unphased and phased reference panels when comparing accuracies between imputed HDRA individuals and their corresponding known genotypes from the 3KRG resequencing panel (Supplementary Fig. 2a). Though there was no loss in accuracy, imputation using unphased reference panels was orders of magnitude slower. We therefore opted to use phased reference panels for the remainder of our imputation testing.

Imputation depends on there being haplotype segments in the reference panel that are similar to those found in the study panel. To explore parameter settings on imputation accuracy, we imputed chromosome 3 in 15 equally sized chunks. We used the full reference panel ($n = 2973$) and varied the parameter $k$, the number of conditioning states chosen for each study haplotype per Monte Carlo Markov chain (MCMC) iteration, at values of 5, 50, 100, 200, and 400 and an effective population size ($N_e$) of 10,000, 20,000, and 30,000 (maximum allowed by IMPUTE2). Previous estimates of effective population size for domesticated rice and its wild relatives ranged from 12,000 to 46,000[21]. Consistent with previous reporting that the IMPUTE2 algorithm was robust to changes in $N_e$[22], we observed no major effect of $N_e$ on chromosome 3 imputation accuracy (site-based $r^2$). There was a slight positive effect of increasing

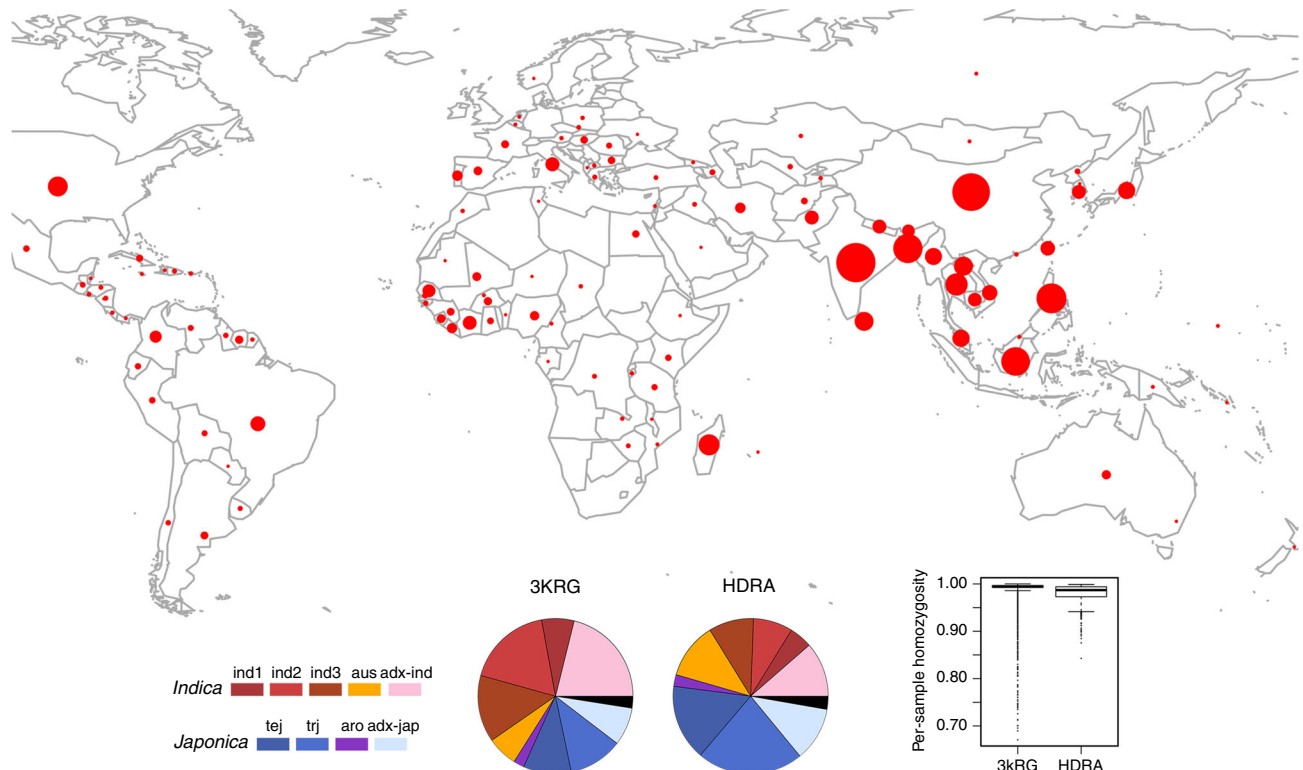

**Fig. 1** *Oryza sativa* diversity panels used in this study. Geographical distribution of the country of origin for the 3KRG (3000 Rice Genomes Project; $n = 3023$) and HDRA (High-Density Rice Array; $n = 1568$) panels. Pie charts depict subpopulation composition of each panel; *ind1, ind2* and *ind3* = geographically differentiated groups within *indica; adx* = admixed within Varietal Group; Black pie segments = admixed across *Indica-Japonica*. Boxplots show distributions of per-sample homozygosity in the 3KRG and HDRA panels (medians at 99.5% and 98.5%, respectively). Boxplot features: thick line = median; box edges = 25th and 75th quantiles; whiskers = lowest and highest observed values within 1.5 interquartile range from 25th and 75th quantiles, respectively

k (Supplementary Fig. 2b–c). Median accuracies for all $N_e$ by $k$ combinations were 1.0, computed for polymorphic markers in the Gold Standard Panel. To assess the effect of reference panel size ($R$), we randomly subsampled the full reference panel ($n = 2973$) at four sizes ($R = 50, 100,$ and $500$) 50 times each and imputed a 2 Mb test region on chromosome 3 (position 7–9 Mb), with fixed $k = 100$ and $N_e = 10,000$. There was a positive effect of increased $R$ on imputation accuracy, especially for markers with low minor allele frequency (MAF) (Supplementary Fig. 2d). We next imputed the full length of chromosomes 1 and 12 using the same gold standard study panel (Supplementary Table 1) to ensure that high accuracies at these selected parameter settings ($k = 100$; $N_e = 10,000$; $R = $ full) were not unique to chromosome 3. The resulting site $r^2$ across chromosomes 1 and 12 was nearly 1.0 for polymorphic markers (Supplementary Table 2). High accuracies were maintained after removing potential close relatives of the Gold Standard Panel from the reference panel prior to imputation (Supplementary Fig. 3).

Because the Gold Standard Panel was largely represented by *tropical japonica* and *indica*, we next checked whether subpopulation-specific limitations to imputation might exist. To do so, we generated subpopulation-specific study panels by randomly sampling ($n = 50$ per study panel) from the 3KRG and imputing them independently using separate reference panels composed of the 3KRG minus the individuals from the study panel of interest, leaving 2973 individuals in each reference panel. The study panels were TEJ, TRJ, ARO, AUS, IND1, IND2, and IND3, representing *temperate japonica, tropical japonica, aromatic, aus*, and three *indica* subgroups, respectively. Three study panels were used for the *indica* subpopulation because the large proportion of *indicas* found in 3KRG (59.7%) resulted in further

stratification of the *indica* subpopulation (Supplementary Fig. 1c). Imputation accuracy across the MAF spectrum was generally high across all seven study panels (Supplementary Fig. 4, Supplementary Fig. 5); however, IND1 was observed to have a lower accuracy than IND2 and IND3 at markers with MAF < 0.20 (Supplementary Fig. 5b–d).

To see whether markers of low imputation accuracy were dispersed or grouped, we examined the distribution of accuracy values across the physical distance of chromosome three for IND1, IND2, and IND3 (Supplementary Fig. 6a). A half dozen clusters of markers with low accuracy were observed in IND1, with half as many observed in IND2 or IND3. Interestingly, two clusters toward the end of the chromosome appear to be shared, one cluster between IND1 and IND2, and a different cluster between IND1 and IND3. Geographical association of the three *indica* subgroups (Supplementary Fig. 6b) and the fact that IND2 and IND3 are more widely dispersed while IND1 is primarily found in North Asia could suggest that genetic architecture related to local adaptation may underlie different imputation performance. IND2 and IND3 accessions were additionally found outside of Asia, as far as South America and Africa. Further analyses would need to be done to document whether a higher frequency of de novo mutations or introgressions (e.g., perhaps from *Japonica* subpopulations) may have given rise to unexpected haplotypes in IND1 and whether these relate to different selection regimes, proximity to wild ancestors, or specific adaptation to northern versus southern climes in Asia.

Using parameter values of $k$ at 100 and $N_e$ of 10,000, we next imputed the full HDRA Panel ($n = 1568$) filtered to 160 K SNPs out to 4.8 M SNPs using the full 3KRG as the reference panel (Table 1, Supplementary Fig. 1). Average concordance as reported

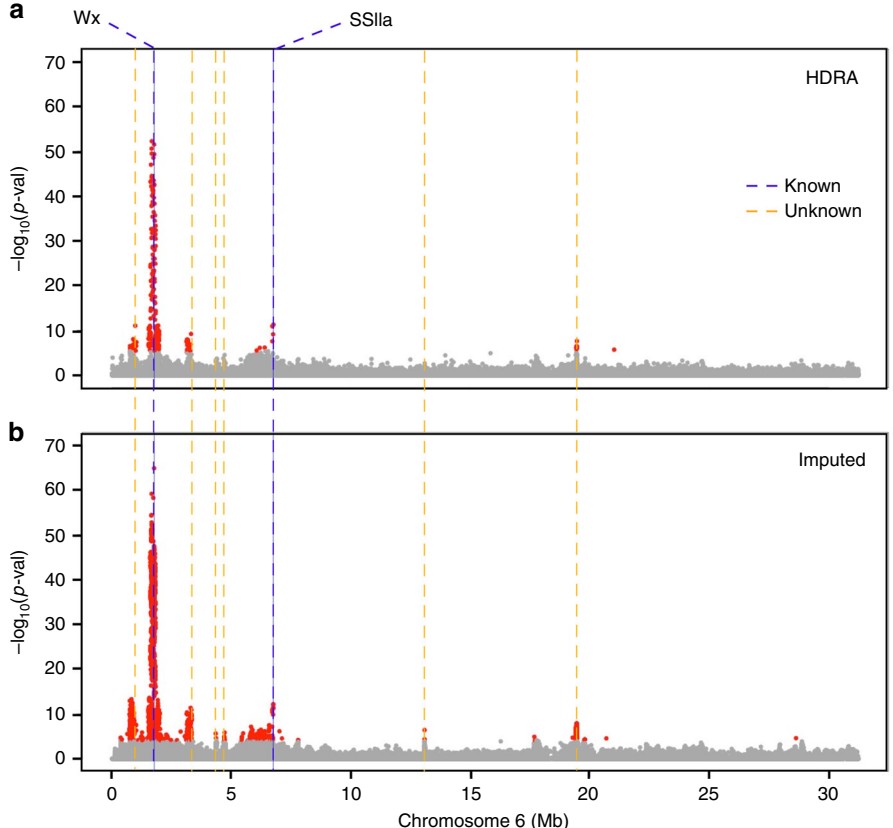

**Fig. 2** GWA study using unimputed and imputed genotype information. Results for grain amylose content in 326 *indica* individuals for chromosome 6. **a** Results using the original genotype data from a 700 K SNP array. **b** Results using imputed 4.8 M SNPs. Dashed blue lines = known genes for amylose content (genes *Wx* and *SSIIa*); dashed orange lines = novel QTL for amylose content

by IMPUTE2 across all 180 imputation runs (12 chromosomes × 15 chunks) was 97.2% (Supplementary Data 2). We next turned our attention to examining the downstream effects of utilizing this imputed data for association analysis.

**Association analysis of grain amylose content**. Signal resolution of genotype–phenotype association benefits from an improved ability to pick up historical recombination breakpoints. In rice, a previous augmentation from 44 K to 700 K genome-wide SNPs in conjunction with an increase in the number of observations (from 413 to 1568 accessions) resulted in markedly improved associations and enabled the approach of subpopulation-specific GWA studies[13,16]. To test whether additional expansion of marker number, without a concurrent increase in panel size, would result in further improvement in genotype–phenotype association signal, we evaluated grain amylose content on 1222 accessions of the HDRA Panel and performed GWA for this trait. Using the original 700 K SNP array data set[13], we analyzed genome-wide scans with all individuals (ALL; $n = 1122$) and also on specific subpopulations classified by the original study[13], *indica* (IND; $n = 326$), *aus* (AUS; $n = 158$), *temperate japonica* (TEJ; $n = 191$), and *tropical japonica* (TRJ; $n = 247$) (Supplementary Fig. 7). We detected strong association peaks from each genome scan except for AUS, for which there were no well-defined peaks. A highly significant region was localized on chromosome six, at ~1.7 Mb and was detected in ALL, IND, TEJ, and TRJ. A second, well-supported peak was also found at ~6.8 Mb in each panel except for TEJ. These two genomic regions harbor known genes that affect amylose content: a major-effect gene, *Wx* (LOC_OS06g04200: position 1,765,622–1,770,656), and *SSIIa* (LOC_Os06g12450: position 6,748,358–6,753,338)[23–27]. We

therefore targeted chromosome six for further study on the potential of imputed data to improve GWA studies or downstream analyses.

**Detection of the primary FNP in *Wx***. Association analysis of chromosome six was next carried out using imputed data for IND. We rationalized that if a sevenfold increase in marker density had a potential benefit to resolving associations in rice, this improvement would be observed in the *indica* clade, as it has the most rapid average LD decay among all subpopulations[13,16,28]. Using the imputed data (imputed chr 6 = 445,466 SNPs), we detected eight peaks for amylose content on chromosome six, including regions containing *Wx* and *SSIIa*. Three of these associations were novel to the high-density data set, i.e., they were not detected using the original unimputed data. For the major association region containing *Wx*, the general width and shape of the peak remained largely the same using imputed data (Fig. 2), indicating that LD, not the number of markers, likely limits the resolution of GWA studies in rice. Estimated local LD value of this associated region (Methods) was 207 kb (Supplementary Fig. 8; Supplementary Table 3). The total LD region encompassed 62 gene models, including transposons and hypothetical proteins (Supplementary Data 3). Despite the apparent limitation of imputed data to resolve associations due to high LD, we detected improvement to the most significant SNP (msSNP) for the *Wx* locus compared to original associations, increasing from $-\log_{10}(p\text{-val})$ of 52.2 to 64.8 (Fig. 3). The msSNP also shifted 96 kb in position, from 1,669,314 to 1,765,761 bp (Fig. 3a, c). This new, imputed msSNP fell within the *Wx* genic region (Fig. 3c) and by cross-referencing it to existing literature, we determined that this G/T msSNP at 1,765,761 matched a previously reported major functional nucleotide

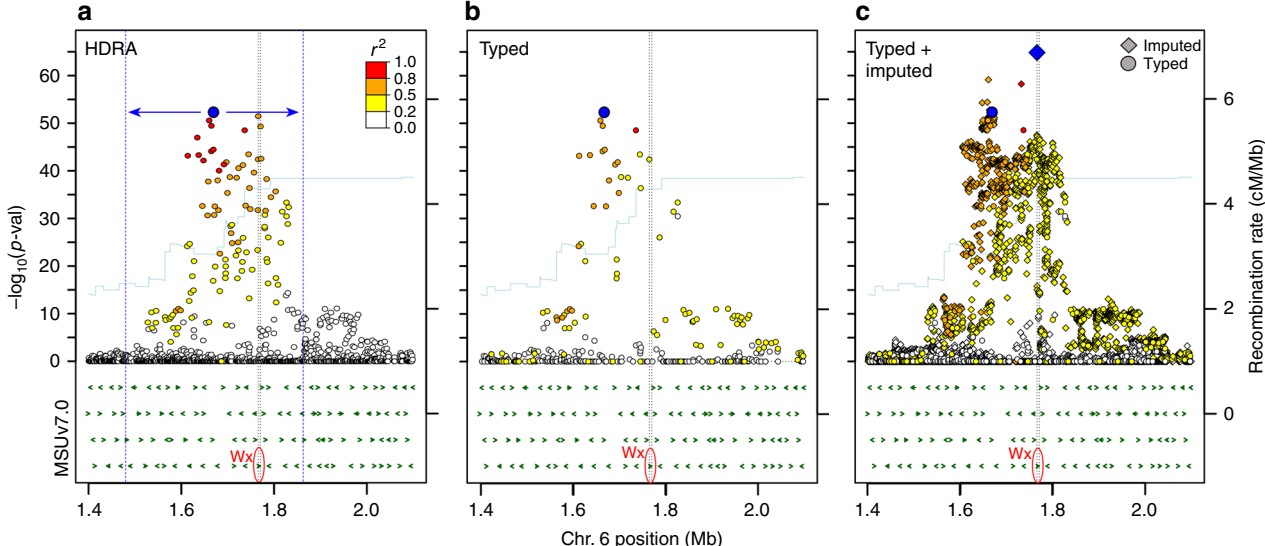

**Fig. 3** Imputed data detect the FNP at major-effect gene, *Wx*. In total, 700 kb regional plots around the *Wx* locus are associated with amylose content. Results are shown for **a** the original HDRA data; **b** association analysis using imputed data with only typed SNPs displayed; and **c** association analysis using imputed data with both typed and imputed SNPs displayed. Colors represent LD ($r^2$) with the msSNPs in the unimputed and imputed analyses, found at positions 1,669,314 and 1,765,761, respectively. The region demarcated with blue arrows and blue dashed lines in **a** represents the extent of local LD of the msSNP (blue circle). Diamonds represent imputed markers, while circles represent typed markers. The blue diamond in **c** represents the msSNP for the imputed association analysis, which matches the known major FNP of *Wx*. In all panels, the start and end positions of *Wx* are marked by dashed black lines. Gene models in this region are annotated by green arrows. Estimation of the recombination rate is displayed in light blue

polymorphism (FNP) in the *Wx* gene[29–31], where AGGTATA versus AGTTATA affects *Wx* transcript splicing and accounts for 80% of amylose content variation in non-waxy rice varieties[32]. Being able to directly interrogate the FNP accounts for the observed improvement in signal strength using imputed data.

Increased SNP density resulting from imputation led to a higher number of markers that fell within gene models. Using the original 700 K SNP data set, there were an average of six SNPs per gene in this ~400-kb region. With the imputed SNP data set, this average increased to 30 SNPs per gene (Supplementary Fig. 9). Based on the original 700 K SNP data set, ten SNPs were found within the *Wx* genic region (chr 6: 1,765,622–1,770,656), only one of which passed the false discovery rate (FDR) threshold. With imputed data, we detected 45 SNPs (Fig. 4a), 22 of which were significantly associated with amylose content in *indica*. Having more significant genic SNPs available is informative when analyzing candidate gene haplotypes, and can be especially useful in cases where an FNP is not imputed directly but can be tracked using haplotypes.

**Gene haplotype analysis of *Wx*.** Using the 22 significant genic SNPs detected within *Wx* for IND, we analyzed gene haplotypes to see if any of them were associated with amylose content. Previous studies had reported low amylose-associated polymorphisms to originate from the *Japonica* clade[33]; we therefore included *temperate japonica* ($n = 191$) and *tropical japonica* ($n = 247$) to help define haplotypes. In total, we identified six major gene haplotypes, WxH.1-H.6, all of which were present in *indica*, although at varying frequencies. The 438 *Japonica* accessions included in the analysis only harbored H.4 and H.5 (Supplementary Fig. 10a; Supplementary Data 4), which were also the lowest-frequency haplotypes in *indica*, suggesting that their presence in the *indica* subpopulation is likely the result of *Japonica* introgression. Meanwhile, H.1, H.2, and H.3 were the most frequent haplotypes in *indica*, together accounting for 84%, and presumed "wild-type" classes. These three groups were associated with the highest amylose content (median~25%; Supplementary

Fig. 10b), consistent with high amylose being the ancestral phenotypic state[33]. H.6 was found only in *indica* and was associated with low amylose content (median < 5%), while H.4 and H.5 displayed intermediate classes of amylose content (median~20 and 14% respectively). H.6 only differed from H.4 and H.5 by one SNP at 1,768,998, so it is likely that H.6 is also the result of *Japonica* introgression into *indica* with subsequent acquisition of the extra variant. This was additionally supported by phylogenetic tree construction using all 45 genic SNPs (22 significantly associated SNPs + remaining 23 nonsignificant SNPs), which showed *indica* H.4, H.5, and H.6 accessions to cluster with the *Japonica* H.4 and H.5 individuals (Supplementary Fig. 10d).

When we conditioned our model on the FNP and re-ran association analysis, we found that most associations disappeared, consistent with the expectation if those markers were in LD with the FNP (Fig. 4b, c). Interestingly, we also discovered a new SNP located at position 1,768,000 bp in the fifth exon of the *Wx* gene that was not significantly associated with amylose content in the original analysis but emerged as significantly associated upon conditioning with the FNP (Fig. 4c).

**Detection of an FNP in *SSIIa*.** The secondary association peak on chromosome 6 contained *SSIIa*, the gene primarily responsible for gelatinization temperature and known to be pleiotropic for amylose content[24–26]. Here, we found only modest improvement in association strength compared to an unimputed GWA study (Fig. 2). The msSNP of this region increased its $-\log_{10}(p\text{-val})$ from 11.3 to 12.2. This small difference is likely due to the fact that the original msSNP from the HDRA data set was already located very close to the known functional gene (Supplementary Fig. 11a), just ~10 kb downstream, and theoretically strongly tagged the underlying FNP. The new msSNP detected using imputed data was found at position 6,752,887, and localized within exon eight of *SSIIa* (Supplementary Fig. 11b–d). From the imputed data set, there were 62 markers within *SSIIa*, two of which were significant for amylose content (Supplementary Fig. 11d). The second of these SNPs had a $-\log_{10}(p$-val) nearly equal to that of the msSNP, at 12.05, and localized at

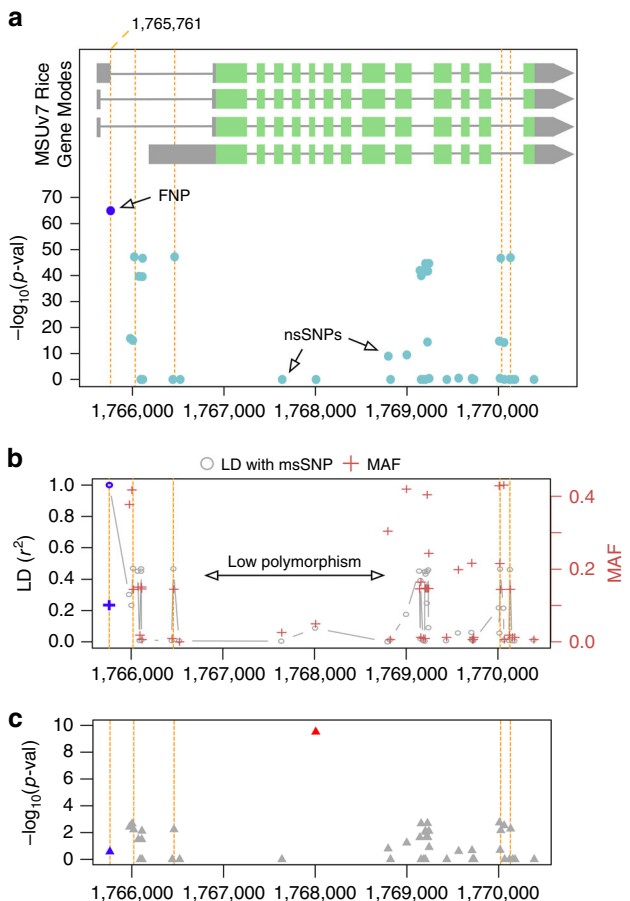

**Fig. 4** Zoom-in plot of *Wx* gene. **a** Association analysis results using imputed data for the genic region of *Wx* (position: 1,765,622–1,770,656). The FNP/msSNP is depicted as a blue circle and the top five most significant SNPs within this region are annotated with orange dashed lines. Two genic SNPs at positions 176,737 and 1,768,998 are non-synonymous SNPs based on the original 4.8 M SNP data set from the 3KRG and are labeled "nsSNPs". **b** LD and MAF of *Wx* genic SNPs. The plot of LD ($r^2$) of each marker with the msSNP (gray open circles) is overlaid with a plot of the minor allele frequency of each marker (maroon crosses). Values of the msSNP itself are highlighted in blue. A region of low polymorphism (i.e., low MAF or few markers) is noted. **c** SNP associations after including the *Wx* FNP as a covariate. Red triangle = new significant SNP at position 1,768,000. Blue triangle = original *Wx* FNP/msSNP

position 6,752,888, just next to the msSNP. Upon comparison to known functional variants of *SSIIa*, we discovered that these two adjacent SNPs matched a previously reported GC/TT polymorphism that explained 62.4% of variation in pasting temperature, another cooking and eating quality trait, which corresponded to an amino acid change at Leu781 of *SSIIa*[34]. In our IND panel, the major haplotype SS.1 (GC; 72% frequency) was associated with higher amylose content and the minor haplotype SS.2 (TT; 28% frequency) was associated with lower amylose content, though distributions were overlapping (Supplementary Fig. 11e), consistent with the relatively minor role reported for *SSIIa* in modulating amylose content. We checked 431 *Japonicas* (185 *temperate* and 246 *tropical japonica*) that had genotype data at this FNP to see if these haplotypes were aligned with either *Japonica* subpopulation. SS.1 was observed in the vast majority of *tropical japonicas* (91% SS.1 and 9% SS.2), while *temperate japonicas* had a random assortment of both haplotypes (49% SS.1 and 51% SS.2).

**The RICE-RP and Rice Imputation Server**. While the original assembly of HDRA aimed to create a germplasm set that was relatively balanced across the five recognized subpopulations, the composition of 3KRG favored sampling of *indica*, which comprised over 50% of the total panel (Fig. 1). To test if adding HDRA samples to the 3KRG data set to create a combined reference panel offered any benefit for rice imputation, we merged the original 700 K data set on HDRA with the 4.8 M data set of 3KRG using IMPUTE2's option for merging reference panels via reciprocal imputation. We then asked whether this combined data set, the RICE-RP, with 4481 unique samples at 5,231,433 markers, could improve imputation of the IND1 subgroup. IND1 was chosen as the study panel here because it had performed the least favorably compared to the other *indica* subgroups and other subpopulations (Supplementary Fig. 4–6) and therefore had room for improvement. For the reference panels, we used RICE-RP (minus IND1 study individuals) and 3KRG (minus IND1 study individuals), and tested them for chromosome three. We observed that at the low end of the MAF spectrum (MAF < 0.158), RICE-RP outperformed 3KRG (Fig. 5a). At the higher end of the MAF spectrum, imputation with RICE-RP resulted in wider accuracy distributions than imputation with 3KRG alone although accuracies were still high (>95% $r^2$) (Fig. 5b). Given that minor allele frequency distribution in IND1, like other rice subpopulations, lays heavily toward the low end of the spectrum (Fig. 5c, Supplementary Fig. 4–5, 79% of the original 3KRG markers on chromosome three had 0.158 MAF), overall performance appears to be improved by the addition of HDRA individuals to the 3KRG.

To see whether rice imputation using the parameters selected in this study for imputing array data was generalizable for sparse data sets, we selected a subset of 16 *tropical japonica* varieties from a previously published GBS data set consisting of 38,618 SNPs[35]. These 16 varieties were chosen because they fulfilled two criteria: (1) they were not part of RICE-RP and (2) they were part of an independent resequencing effort[15]. We used the resequencing data set from Duitama et al. as validation genotypes for assessing imputation accuracy. In total, 1.2 M SNPs in the resulting imputed data set could be used for accuracy ($r^2$) computation because they were polymorphic in the 16 varieties and were also typed in the resequencing data (Supplementary Fig. 12). Genome-wide median accuracy was 1.0, consistent with the high imputation accuracy of *tropical japonica* shown in Supplementary Fig. 4b.

To extend the capability of performing imputation out to the greater rice research and breeding community, we developed a web-based application called the Rice Imputation Server that utilizes IMPUTE2 at the backend (Fig. 6). This publicly available service allows users to upload their own data, e.g., genome-wide SNPs generated by GBS methods, and receive imputed data sets back. We integrated the 3KRG and HDRA Panels as a single, phased RICE-RP at 5,231,433 SNPs via reciprocal imputation and phasing and provided four "SNP filters" to facilitate downstream trimming of imputed data sets, which may be desirable depending on the end application. These SNP filters represent the set of (1) genic SNPs, (2) exonic SNPs, and (3) putative splice sites. Other options for filtering include basic Plink1.9 utilities such as LD-based pruning and random thinning, which the user may also opt to perform after imputation (Fig. 6). This service is available at http://rice-impute.biotech.cornell.edu.

## Discussion

In this study, we demonstrate the high accuracy of imputation for rice genetic data using IMPUTE2. We first determined optimal

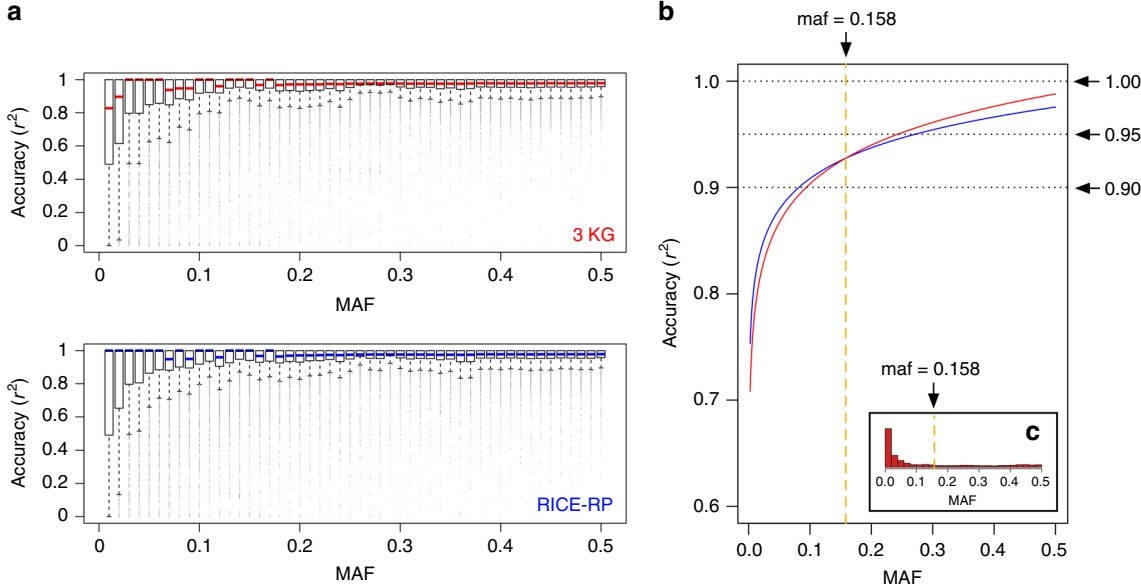

**Fig. 5** Comparison of imputation results using RICE-RP versus 3KRG as a reference panel. **a** Imputation accuracies across minor allele frequency (MAF) bins of size 0.01 for sites that were polymorphic in both imputed data sets for IND1 subpopulation chromosome three. **b** Fitted logarithmic curves for imputation accuracy across the minor allele frequency spectrum in IND1. Using RICE-RP as the reference panel, it performs slightly better than 3KG at low MAF (<0.158) and slightly worse at higher MAF. **c** Distribution of MAF in IND1, showing that relatively few markers have MAF of greater than 0.158 (larger version found in Supplementary Fig. 5b)

parameters for rice imputation using a gold standard set of germplasm with known "true genotypes" and then imputed the entire HDRA Panel of accessions out to >4.8 M markers. This work elevates the utility of a previously released genetic data set that is associated with publicly available, purified genetic stocks designed for GWA studies in rice[13]. Using the unimputed and imputed data sets, we compared association analysis results for amylose content, a critical eating and cooking quality trait in rice[36]. Although addition of markers cannot overcome the resolution deficiencies caused by naturally high LD in rice populations, increasing marker density creates a greater opportunity to detect causal SNPs or functional polymorphisms responsible for phenotypic variation. This in turn boosts signal strength of the msSNPs at association peaks.

Having genome-wide imputed SNP information across the HDRA Panel can facilitate population genetic analyses on genes or regions of interest identified by GWA studies, without the need for additional sample sequencing. Using the set of significant genic SNPs in *Wx* from the imputed data set, we generated gene haplotypes that resolved all four classic phenotypic groupings for amylose content in *indica* rice and performed an allele-specific Extended Haplotype Homozygosity analysis across a 200-kb region containing the *Wx* gene using the msSNP as the focal SNP (Supplementary Fig. 10c). Haplotypes that strongly associate with traits such as grain quality are critically valuable to rice breeders, who can use this information to make selections even if the FNP itself is unknown in their material. These types of population genetic analyses are not possible using the original, unimputed HDRA data set due to the ascertainment bias inherent in the array design[13], e.g., selection for non-synonymous SNPs.

One limitation to the data sets (Table 1) developed in this study is that variants present depend on SNPs found in the 3KRG 4.8 M data set, which had been originally filtered for 0.01 global minor allele frequency. This dependency can lead to the potential of missing rare variants in the final imputed data set. Rare SNPs, e.g., those private to a specific subpopulation at low frequency that were filtered out of the 3KRG 4.8 M SNP set would be absent. It would be of interest in the future to generate an updated RICE-

RP derived from an unfiltered initial reference panel, or one that accounted for subpopulation-specific SNP frequency information. Such strategies would help retain rare variants in final imputed data sets.

One interesting insight that emerged from investigating subpopulation-specific imputation accuracy is that imputation may offer a means to identify gaps in gene bank collections with respect to geographic (and eco-climactic) variation and to rationally sample additional accessions for further resequencing. In our study, we observed lower imputation accuracies in IND1, a subgroup of the *indica* subpopulation that is localized in China, suggesting that data gaps exist for this subgroup. Passport information, which includes collection location (longitude and latitude), is available for about 50% of gene bank samples and could be used to resolve location data for those that only have country-level information in a manner similar to that of Elhaik et al[37]. With the imputed location data, the Focused Identification of Germplasm Samples[38] could be invoked with new samples chosen to fill gaps in sampling across eco-regional and climactic clines. If a locale already has several accessions that have been resequenced at high coverage, then new accessions from the same area or from a similar eco-climactic zone could be resequenced at lower depth. For accessions that fill prospective gaps in the collection, deeper resequencing would likely be beneficial, so that novel, rare alleles may be identified. These approaches can therefore assist gene banks in extending genotypic data to optimally cover the species diversity, allowing for heightened value and better use of resources.

Having the ability to efficiently impute rice genetic data opens the door to a wide range of applications that extend beyond the few demonstrated in this study. Perhaps of the greatest immediate impact, it facilitates integration of data sets. Crop researchers commonly handle a range of germplasm types, resulting in genomic data sets of varying marker densities on independent collections of samples, each developed to address specific problems of interest. Imputation facilitates the integration of these resources by mapping them onto a common genomic framework without the added time and expense of re-genotyping. It also

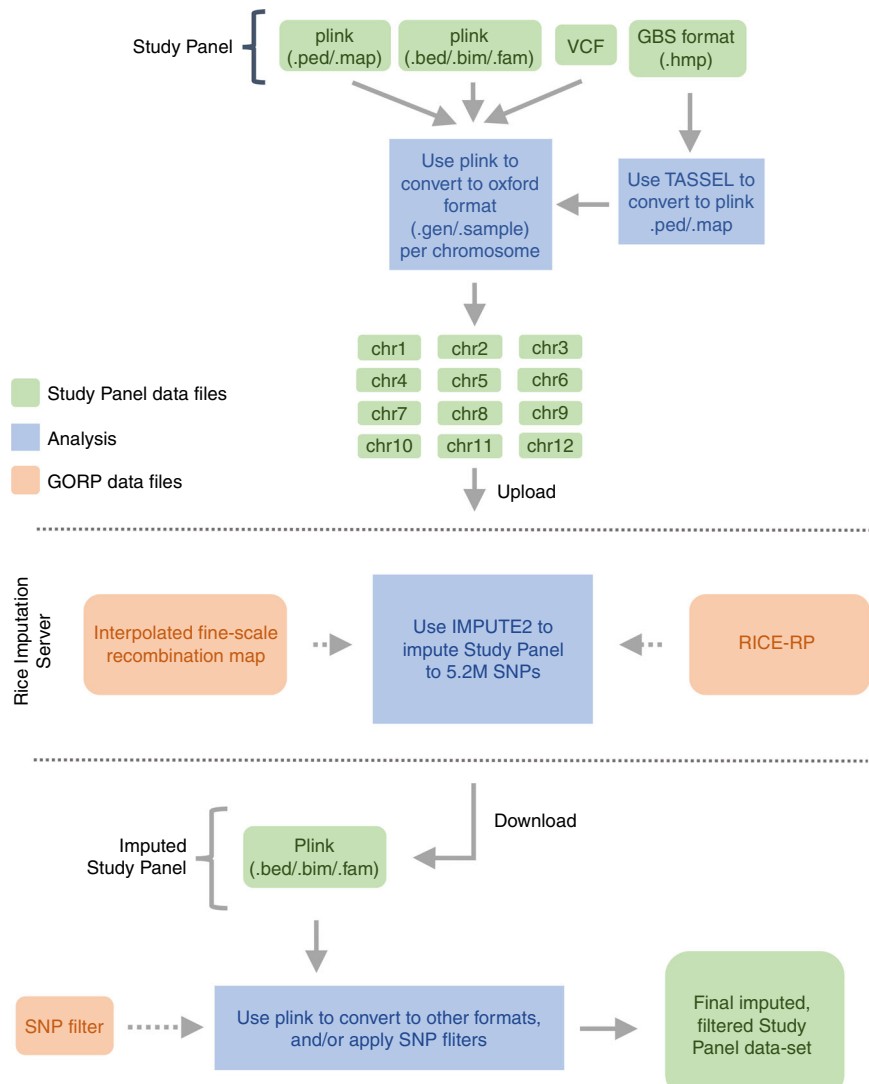

**Fig. 6** Steps for imputing rice genetic data using the Rice Imputation Server. The user uploads input files in oxford format (.gen/.sample) per chromosome. The imputation server utilizes chromosome recombination maps and the Rice Reference Panel (RICE-RP) haplotypes to impute the user's data out to 5.2 M SNPs with IMPUTE2 and returns imputed results in plink binary format. The user can filter the imputed data using plink1.9 to produce a final data set of desired SNP density and composition

offers the option of waiting to make a final selection of germplasm or SNPs until after integration, tailoring the best selection possible for the researcher's need. Imputation provides the means to update previously genotyped panels quickly and at little cost, often dramatically increasing the resolution of existing resources. As GWA experiments in rice continue to accumulate in the literature, imputation can enable future GWA meta-analysis[39] on agronomically critical traits by allowing the combination of germplasm and phenotypes across published studies.

Increasing access to imputation tools is one measure to encourage interoperability of crop genomic resources, promote re-use of publicly available and internally collected data, augment the impact of individual data sets, and inform future gene bank resequencing efforts. To facilitate imputation analysis for rice researchers, we developed a publicly available imputation web tool called Rice Imputation Server. This service harnesses the results of this study and leverages haplotype diversity found within the Rice Reference Panel. One critical feature that will likely contribute to long-term utility of this tool is its modularity; it is set up to readily accept upgrades to either the imputation software (e.g., a new version of IMPUTE2) or data (e.g., a higher

density version of RICE-RP) as they develop. Future directions for improvements to the service may include offering region-specific imputation for researchers interested in data at certain loci of interest rather than genome-wide and incorporating indel or other non-SNP variant imputation capacities. To help users manage their resulting large data bundle, we offer "SNP filter" text files that can feed directly into Plink1.9 along with the user's imputed plink binary data set as a functional approach to filtering genome-wide SNPs. Making imputation capability more accessible to plant research communities may improve the power of phenotype–genotype association studies, enhancing the productivity of basic investigations into molecular mechanisms and simultaneously accelerating translation into productive, stress-tolerant, and nutritional plant varieties that are urgently needed to enhance global food security.

## Methods
**Genotype information**. SNP information on 1568 individuals genotyped on the High-Density Rice Array[13] and 3024 individuals genotyped as part of the 3000 Rice Genomes Project[14] were previously published and now have been assigned germplasm doi's for the first time (Supplementary Data 1). These two germplasm

panels are hereafter referred to as the HDRA and 3KRG Panels, respectively. An intersecting set of markers between the 18 M SNP resequencing data set (http://snp-seek.irri.org/_download.zul) and the 700 K SNP array data set were filtered for a maximum of 5% missing data per marker in each of the two panels and a minimum of 1% minor allele frequency across the combined data set. The resultant set of ~160 K high-quality SNPs was used in imputation analyses (position list of 160 K SNPs can be found at SNP Seek; http://snp-seek.irri.org/_download.zul). Population structure was analyzed on the 4591 combined sample set using *fastStructure* and the cut-off used to assign samples to specific subpopulations was 75% ancestry. One 3KRG accession (IRIS_313–8921) was dropped from this study due to excessive missing data at the 160 K SNPs. The filtered, 4.8 M SNP data set from 3KRG Release 1.0 was used as the high-density reference set of SNPs and was selected due to its filter on missing data (max 0.2). By design, the 4.8 M SNP data set had been filtered for 0.01 global minor allele frequency in the 3KRG Panel. An overview of data sets used in this study is found in Table 1 and Supplementary Fig. 1.

**Phenotype data**. Samples were grown during the 2011 wet season at the International Rice Research Institute in the Philippines using standard crop management practices for irrigation and pest control. Grain was harvested at maturity and stored for 6 weeks under controlled relative humidity to equilibrate for moisture content. In total, 10 g samples of paddy rice were dehulled and polished by placing grains into 10 mL capsules with fused aluminum oxide and abrading gently for 1 h in a paint shaker. Twenty whole-milled rice kernels were ground in a Udy cyclone mill (sieve mesh size 60), 100 mg of rice powder were weighed into a 100 mL volumetric flask, and 1 mL of 95% ethanol and 9 mL of 1 M sodium hydroxide are added. Amylose content was measured using the standard iodine colorimetric method ISO 6647-2-2011[40]. Absorbance of the solution was measured using an Auto Analyzer 3 (Bran + Luebbe, Norderstedt, Germany) at 600 nm. Amylose content was quantified from a standard curve generated from absorbance values of four standard rice varieties (IR65, IR24, IR64, and IR8).

**Imputation and RICE-RP assembly**. IMPUTE2[41] and SHAPEIT[42] were used for all imputation and phasing activities, respectively. Recombination rate across the rice genome was estimated using genetic data from a Recombinant Inbred Line population[43] and mean-adjusted to account for varying reported values for genome-wide recombination rate average[44–47]; this was used as the fine-scale recombination map input for imputation. To assess the performance of IMPUTE2 on rice data, we tested different imputation parameter settings on a "gold standard" study panel containing 50 accessions that overlapped between the 3KRG and HDRA panels (Supplementary Table 1). The reference panel use for parameter testing was the 3KRG Panel minus the 50 accessions that were part of the gold standard study panel ($n = 2973$). We tested combinations of the following parameter values/settings: phased versus unphased reference panels, size of the original reference panel ($n = 100, 500$), effective population size ($N_e = 10,000, 20,000,$ and $30,000$), and number of reference individuals sampled per MCMC iteration of the imputation to use per study individual ($k = 5, 50, 100,$ and $200$). As recommended by IMPUTE2 developers, imputation was performed in small chunks per chromosome by dividing each chromosome into 15 equally sized regions ranging from 1.5 to 2.88 Mb depending on chromosome size, and parallelized across 15 cores per chromosome. The resultant imputed chunks were merged to produce one file per chromosome. The final selected parameters for imputation of the entire HDRA Panel were $k = 100$, $N_e = 10000$, reference panel = phased, reference panel size = full 3KRG Panel (minus one individual removed due to 98% missing data at the 160 K SNP data set), and imputation chunks per chromosome = 15. Imputation accuracy throughout this study was assessed as marker-based $r^2$ between imputed genotypes (after converting dosage.gen files from IMPUTE2 to genotypes in plink format via plink1.9) and true genotypes. To assemble the RICE-RP, we used the merge reference panel option of IMPUTE2 with the HDRA Panel at 700 K SNPs and the 3KRG Panel at 4.8 M SNPs. Merging via reciprocal imputation was performed in ten chunks per chromosome, using the same parameters as for imputation: $k = 100$; $N_e = 10000$.

**Association analysis**. Using the 700 K SNP array data set[13], genome-wide scans for amylose content were analyzed using all phenotyped individuals (ALL; $n = 1122$) as well as on individual subpopulations, *indica* (IND; $n = 326$), *aus* (AUS; $n = 158$), *temperate japonica* (TEJ; $n = 191$), and *tropical japonica* (TRJ; $n = 247$) (Supplementary Fig. 6). Using the imputed data set on the HDRA Panel, association analysis was performed for chromosome six in IND. All associations were analyzed using a linear mixed model implemented by the gwas() function within the rrBLUP R package[48]. Parameters for the gwas() function are as follows: min.MAF = 0.05, P3D = TRUE, and K = a kinship matrix. The kinship matrix was computed from the original HDRA array data using the A.mat() function in rrBLUP[48]. Three additional principal components (PC)s were included for the ALL group to control for high-level stratification, while no additional PC covariates were included for subpopulation-specific analyses, following the approach used in McCouch et al.[13] (Supplementary Fig. 13). Manhattan plots were generated using the qqman R package[49] and a Benjamini–Hochberg false discovery rate was set at 1%[50] to deal with multiple testing. Regional plots were generated using a code

based on that provided by the Diabetes Genetics Initiative of the Broad Institute. All gene annotations used here were taken from the MSUv7 genome assembly (http://rice.plantbiology.msu.edu/). Local LD associated with significant SNPs was determined following the method described previously[35].

**Extended haplotype homozygosity**. EHH analysis (Supplementary Fig. 9c) was performed for the msSNP of the *Wx* gene using the R package rehh[51]. A 200-kb region flanking each side of the focal SNP was used.

**Rice Imputation Server**. The Rice Imputation Server takes in rice genotypic data sets from public users, runs steps for data re-formatting and imputation remotely, and returns imputed data back to users. To accommodate the large number of parallel tasks that accrue across potential users and per imputation job (12 chromosomes × 10 chunks/chromosome), a Celery queueing framework was implemented for task management. A Graphical User Interface was developed as a web application to facilitate data set exchange. The Rice Imputation Server can be accessed at http://rice-impute.biotech.cornell.edu.

**Code availability**. The script used for running GWA using the imputed HDRA data set is available at http://ricediversity.org.

**Data availability**. The full RICE-RP dataset is available at the McCouch group site, Rice Diversity (ricediversity.org), the SNP-Seek database (http://snp-seek.irri.org/download.zul), and European Variant Archive (Project accession: PRJEB26328 (https://www.ebi.ac.uk/ena/data/view/PRJEB26328)). Imputed data on the HDRA Panel may be obtained by subsetting the RICE-RP. The imputed *indica* dataset used in our GWA study (based on imputation from the 160K SNPs), along with the kinship matrix, amylose content dataset, and R script used to run the analysis may be accessed at Rice Diversity.

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

## Acknowledgements

We acknowledge the Secretariat of the International Treaty for Plant Genetic Resources for Food and Agriculture (ITPGRFA) for issuing DOIs as unique permanent identifiers for the germplasm resources used to generate the RICE-RP, Teody Atienza and Venea Dara Daygon for collecting grain quality phenotype data, and Jeanne Kisacky for assistance with the formatting of figures and manuscript submission.

## Author contributions

D.R.W. designed and performed experiments, analyzed data, prepared figures, oversaw web application design, and wrote the manuscript. F.J.A.-P. managed data, designed, and implemented the web application. D.C. contributed analyses to the study. Y.S. assisted in database management and contributed ideas to web application design. J.M. contributed ideas to web application design and the manuscript. M.F. collected phenotype data. K.L.M. contributed ideas to the overall study and helped revise the manuscript. N.A. contributed ideas to the overall study. S.R.M. managed the overall study, contributed analysis, and revised the manuscript.

## Additional information

**Competing interests:** The authors declare no competing interests.

