## [Peer Review File · Nature Communications]

Reviewers' comments:

Reviewer #1 (Remarks to the Author):

This manuscript presents the Rice Imputation Server, which is a publically available web application allowing researchers to access a globally diverse rice reference panel to impute genetic information. The claim is that by providing this resource, researchers will be able to capitalize on increased marker density, combine studies more easily, and ultimately identify more precisely polymorphisms relevant for valuable agronomic traits. These claims are well supported from other species such as humans, where imputation has been a standard approach in GWAS for roughly a decade. In concept, the server would be a valuable resource for the rice community; however, there are a number of areas in which additional testing, discussion and guidance would be likely to increase the uptake of this application and hence have greater influence on the field. In particular,

- There is very little discussion or analysis of the effect of having unbalanced distributions of the different subpopulations in the reference panel, nor are the statistics on the makeup of the panel given (though they are probably in prior publications). While the authors mention that this can lead to rare alleles being omitted due to the filtering criteria, it would be good to know whether we expect different imputation efficiencies across the subpopulations. Also, what is the makeup of the gold standard panel of 50 lines in both 3gRK and HDRA? Is the 98% estimate of imputation accuracy based solely on one or two subpopulations (indica/japonica)? This information would be important to understanding the limitations of the imputation.

- On a similar note, there is no discussion of the effect of relatedness in the reference panel and/or the data to be imputed. In human studies the reference panels are unrelated, but this is not necessarily true in crops, particularly since the authors do mention the presence of biological replicates in the panels. In particular, regarding the biological replicates – why are they retained? For the gold standard panel, there is hopefully no overlap, but how closely related are they to the rest of the panel (as this may effect the imputation estimates as well)?

- In terms of generalizability, the authors mention that the panel can be used to impute GBS-based data, although the example presented is for a SNP array. SNP calling in GBS requires alignment to a reference, and can be more error-prone than SNP arrays. What effect does that have on imputation? Do parameters need to be adjusted? Additional analysis/discussion of this point would be needed before making claims about effectiveness in other data types.

- Finally, I am a little concerned about the usage of GORP for imputation since this is already based on one round of imputation (~11K SNPs from 3gRK and 4M+ for HDRA). What effect does this compounding of uncertainty have? Including a comparison of imputation just from 3gRK (which may have lower diversity) vs. imputation from GORP (which has this added level of uncertainty) would help define where potential issues may arise and give more confidence in the approach.

The statistical approaches and packages used for analysis are appropriate; however, not all parameters are given for the analysis (e.g., ll. 481-482 only states what package was used for analysis – rrBLUP – without providing any information regarding functions or parameters). This will make reproducing the work more difficult. Including R scripts as supplementary files would help to alleviate this issue.

Minor comments:

- Can the framework be used for other crops, given an appropriate reference panel? How modular/customizable is this?

There were several small typos:

- L462 use -> used
- L165 Average concordance – average or median Previously used median
- Table 2 – are that many digits needed?
- L75 last sentence does not make sense

Reviewer #2 (Remarks to the Author):

In this work, Wang et al. developed a web-based application for imputation analysis in rice GWAS, based on the software IMPUTE2 and the reference panel (e.g., 3000 rice genomes). The application may be useful for association analysis in rice. However, the paper is obviously lack of novelty since the software and the rice reference panel have been published and widely used. Moreover, I don't think the imputation is a key step in rice GWAS. In fact, the genotyping and follow-up procedures in GWAS (from raw sequence data to high-quality SNP matrix, and from association analysis with mixed model to the identification of candidate genes) include many steps, most of which are more computationally intensive and complicated than the imputation step. Below I have several minor suggestions:

- 1) Why the authors chose IMPUTE2 for imputation? What about other imputation software (e.g., BEAGLE), and what about the situation for heterozygous genotypes?
- 2) Rice has a slow LD decay rate. More examples could be provided in the paper (by the way, SSIIa is also a major gene, not minor effect), where the authors need to focus on the analysis of the causative genes/variants (e.g., the prediction of the effects for gene coding) through the integrations of more variation information from imputations.
- 3) The result section include many technical details (e.g., the parameter settings of the imputation software), which should be moved into the method section.
- 4) I did not find the legend for Fig. 5D.

Reviewer #3 (Remarks to the Author):

This manuscript describes a public server the authors have built for imputing rice genomes. Additionally, it presents benchmarks of rice imputation accuracy (which help identify appropriate parameter choices for imputation using IMPUTE2), and it also showcases examples demonstrating how imputation can be used to refine rice GWAS associations. The imputation server has a user-friendly interface that is well-documented, so there is a good chance researchers will find it handy.

On the whole, the paper is well-written and does a good job of explaining the utility of the server. I do have several (hopefully constructive) criticisms of the paper that the authors should be able to address as well as some minor suggested improvements.

MAJOR COMMENTS:

1. The authors use concordance between held-out and imputed genotypes to measure imputation accuracy (line 165 and lines 476-477). This metric is generally not very informative because (i) it conflates imputation accuracy across the MAF spectrum and (ii) it awards very high accuracy to rare SNPs under any reasonable imputation algorithm (e.g., always imputing the major allele would achieve genotype concordance close to 1). The authors should instead use the standard approach to benchmarking imputation accuracy, which is to measure R^2 between held-out and imputed genotypes as a function of MAF. (Lower-frequency SNPs are harder to impute and have lower imputation R^2 , so most papers plot a curve of R^2 vs. MAF.)

2. The authors selected a gold standard set of 50 samples based on overlap between HDRA and 3kRG. This choice is actually problematic because only two subpopulations (tropical-japonica and indica) are well-represented among the 50 samples. Fortunately, there is really no need to select samples present in both panels. The authors can create a test set from any subset of 3kRG samples by restricting the test set to the 160K filtered HDRA markers, imputing remaining SNPs (using all 3kRG samples not in the test set), and then checking imputation accuracy against the held-out genotypes in 3kRG for the test samples. The authors should perform such an analysis for each of the five subpopulations to determine how well imputation performs on each subpopulation. (Most likely, imputation is harder on AUS.)

3. Does the combined GORP panel actually add any value for imputation vs. simply using 3kRG? I suspect it doesn't, because the imputed HDRA haplotypes that were added into GORP represent very little additional information: the imputed haplotypes have essentially just been copied over from best matches in 3kRG. The authors should check whether GORP actually provides any improvement in imputation accuracy. They could do so quite easily by comparing imputation accuracy on their 50-sample study set using the GORP reference (minus those 50 samples) vs. the benchmark they already performed using 2973 3kRG samples.

4. I'm not sure what to make of the haplotype analysis starting at line 217. Generally, haplotype analysis of greatest interest when the true causal variant is untyped and

unimputed -- but here, there is a clear causal variant in the region (the intron1 splice variant) and as far as I can tell, that variant may explain the entire association signal. A conditional analysis is really needed here: the authors note that 22 SNPs in the Wx genic region significantly associated with amylose content, but before embarking on haplotype analysis, how many of those associations are simply explained by LD to the msSNP? To get at this question, the authors can simply condition on the lead SNP and see how many associations disappear (with the caveat that the lead SNP is likely to be imperfectly imputed, so remaining associations still need to be interpreted with care).

My guess is that many of the 22 associations will turn out to be explained by LD. For example, the authors claim that the second FNP at 1,768,998 is "associated with amylose content" (line 251), but based on $r^2 \sim 0.18$ (Supp Fig 7) to the lead FNP which has $-\log_{10}p = 64.8$, I would expect a $-\log_{10}p$ of $0.18 * 64.8 = 11.7$, which looks to be exactly what is observed in Figure 4 at the second FNP -- suggesting that in fact this signal is just LD.

MINOR COMMENTS:

Table 1: "Haplotype number" is confusing because the genomes are diploid. How about "Unique samples" instead?

Table 1: Why does GORP have a larger number of markers than 3kRG?

Line 117: "O. sativa is an inbreeding species such that genotypes are generally already equivalent to haplotypes" -- quantification of this statement would be helpful (e.g., reporting median and range of per-sample heterozygosity, or better yet, a box plot).

Line 133: I was surprised that imputation accuracy appeared to max out at $R=1500$. In human data sets, accuracy always steadily increases with reference panel size. Does this result still hold after measuring accuracy using R^2 and stratifying by MAF? If so, can the authors suggest why?

Line 165: It's unclear to me what samples were used to measure "average concordance": the 50 study panel samples? (Again, MAF-stratified R^2 would be much more informative here.)

Lines 200-201: "LD, not the number of markers, likely limits the resolution of GWAS in rice" and 317-319: "extensive linkage disequilibrium is the primary constraint to narrowing down association regions rather than the number of markers per se." I think these statements are unnecessarily negative; they seem to argue against the utility of the imputation server! Presumably the authors are referring to the fact that the width of the association peaks remain the same after imputation (as they must). However, in the best case scenario -- which the authors demonstrate here -- imputation can still help fine-map association regions when the true causal variant is successfully imputed and becomes the most significant association (as the authors show at Wx and SSIa; Figure 3 is very nice).

Figure 4: Adding Supplementary Figure 7 (LD and MAF) as a bottom panel of this figure would greatly help its interpretation.

Line 255: Supp Fig 7 should be referenced here instead of Supp Fig 6.

Figure 5: Caption for panel D is missing.

Line 282: What's the LD between the two adjacent SNPs?

Discussion: I would recommend shortening the Discussion.

Line 487-488: Why were no PCs necessary for the subpopulation-specific analyses? The top PCs appear to be explaining large amounts of genetic variance (>5%), assuming that's what the y-axes of Supp Fig 6 are measuring. (The meaning of the y-axes should be clarified in Supp Fig 6.)

Reviewer #4 (Remarks to the Author):

In this work, the authors present the Rice Imputation Server, a publically available web server that provides imputation resources to the rice genetics community. The goal of the paper is to facilitate genetics research in rice by providing a resource for imputation of rice accessions using a globally diverse reference panel. They describe a combined panel made of two sub-reference panels, one the HDRA panel from a 770K array using approx 160K filtered high quality SNPs and approx 1500 samples, and the 3kRG panel from 29M to 4.8M filtered high quality SNPs and approx 3000 samples. The paper is extremely well written and well presented and most analyses are appropriately thorough. The benefit for downstream interpretation of imputation is particularly well presented (e.g. Fig 3).

My two biggest worries about this paper are first, is it really true that high computational demand is limiting uptake of imputation in rice, necessitating this server as stated in the abstract (vs a download link for the reference panels), and second, is imputation from arrays or GBS the best long term strategy for genetics research in rice (and hence, how long will this server truly benefit the community)? On the first point, my understanding with human data is that human phasing and imputation servers arose at least partially out of an inability to release the full HRC reference panel data due to privacy / consent reasons, so imputation servers are the only way for human genetics researchers to obtain the full utility of the complete HRC panel. I assume no such restriction holds here, and the reference panel sizes proposed here are comparable to those used in human genetics research from ~2009-2015 that were imputed by hundreds (thousands?) of groups worldwide, and would not strike me as beyond the capability of a normal lab that sought to perform GWAS. On the second point, either a) you're interesting in prediction, so you don't care what the causal SNP is, so you can use a low density marker array and don't really need imputation, or b) you're interested in the biology. For b), you really want high sensitivity to detect variants, to make sure you've captured the causal variant (FNP). The rice genome is comparatively small (430 Mbp), which makes resequencing an attractive option for genotyping. If I were a

researcher today and wanted to perform GWAS and fine-mapping in 1,000 rice samples from a single population tomorrow, would I choose arrays and imputation? If making libraries can be done for <20USD, and if we believe 100 Gbp / 1000 USD (human 30X for 1000 USD), then 1 Gbp / 10 USD, so you can get a little over 2X low coverage for 30 USD (so maybe 60 dollars at scale in a commercial lab) and you can use Beagle / STITCH / etc for imputation without a reference panel and get high quality genotypes for all SNPs. Are arrays (at the density cited here of 700K?) and reference panels cost and time-competitive and the way forward?

I also have several questions and comments about the paper as presently written.

Major comments:

1) In Results section "Parameter selection for rice imputation", you are selecting parameters by using HDRA samples with imputation from a 3kRG reference panel. But the whole point of the paper is providing an imputation server with a reference panel, and that panel uses HDRA and 3kRG together. Obviously HDRA + 3kRG shouldn't be worse than 3kRG, but is it any better?

To verify this, I would be interested in knowing a) does adding HDRA to 3kRG improve imputation, and b) does this result hold if you remove members from that sub-population from within HDRA? Specifically, I would like to see, for each of IND, AUS, TEJ and TRJ the imputation of the members of the overlap from that subpopulation, using a combined reference panel of (3kRG minus overlap) + (HDRA minus overlap minus subpopulation), as compared to (3kRG minus overlap).

This might be too exhausting for all the parameter benchmarking, but I would do this with final parameters somewhere around where you present Table 2.

Also, just to be clear, other than identical samples, are there any other relatives between the overlap set and either HDRA and 3kRG, with long tracts of shared IBD that would artificially inflate imputation accuracy estimation?

2) The discussion ~340-374 worries me. First, can the authors (somewhere in the paper) give a slightly more detailed breakdown on how 700K -> 160K in the context of the 29M / 19M / 4.8M SNPs from the 3kRG? How many of the array SNPs are removed for frequency, quality or intersection reasons? Also, I don't really understand why the requirement to have >1% frequency in the GORP exists, given you've shown this to be a problem in the Discussion? Why not fix this now instead of later by considering >4.8M SNPs (at least extend 4.8M to include >1% MAF any sub-population to prevent this specific problem)? Finally, this makes me question how time saving an imputation server will be, if follow up work requires downloading all the reference panels to your computer anyway to look at subpopulation specific haplotypes, how much effort are you really saving by performing a first imputation in the cloud.

3) Can you please switch accuracy to per-site squared correlation (methods 475-477, Figure 1, etc). Correlations are much more informative for interpreting power, and it can be hard to interpret accuracy without precise knowledge of the allele frequency spectrum (e.g. if you imputed a dataset with only 1% MAF SNPs completely at random your accuracy would be

approx 98% but your imputed dataset would be completely useless)

4) It seems that a previous imputation server (the Michigan human one) also provides an open source framework for building imputation servers

<https://www.nature.com/articles/ng.3656>

<https://imputationserver.sph.umich.edu/start.html> (this website has a part saying "Build your own imputation server")

Even if this doesn't end up in the paper, I'd be curious to hear a little more about why the authors chose to go with their own server vs build off someone else's work

Minor comments:

Reading Table 1 and lines 300-302, it would seem that 3kRG was imputed into HRDA to create a haplotype reference panel with 4.829M SNPs with 4591 samples (4481 haplotypes). But reading the Methods 477-480 and the IMPUTE2 website, it seems the "merge reference panel" option is used which keeps the two panels separate. Which one does the server actually use? On the server is a single reference panel used or two of them? If a single reference panel, can the authors explain somewhere in a sentence why (does this make subsequent imputation faster? More accuracy?) and specifically how they they do this (i.e. X HDRA SNPs into Y 3kRG SNPs with parameters W, Z, etc)

Table 2 could probably go to the supplement

My imputation job REC_V54VZP of the test data failed with Status: [u'FAILURE', u'PENDING', u'SUCCESS']. Better error reporting is needed (and maybe remove the "u"'s from your JSON / YAML), and obviously a paper on Rice Imputation Server needs to be able to actually impute rice data! I also uploaded a PDF (Status REC_V55VP8 [u'PROGRESS', u'PENDING', u'SUCCESS']) and during the several hours I spent reviewing this paper, the job never failed. It would be extremely useful for users if you could validate all your inputs quickly upon submission (is it a tarball? Does it contain the right number of files? Are the file extensions correct? Do the files conform to input requirements?), as I think anyone who has spent time in bioinformatics will know how much time is wasted debugging discrepancies between the input data that has been prepared and what the program accepts. Centralization of this process would actually be a major plus of this server (centralized debugging)

Is the recombination rate in Fig 3 as expected? From 1.8 to 2.0 Mbp, recombination rate is >4 cM/Mb? This seems implausible in the context of the discussion talking about high LD in these mapping populations, and Fig 3C which suggests recomb hotspots at ~1.55, 1.62, 1.8 and 1.98 Mbp. I would love it if my mapping species all had uniform recombination rates of >4 cM/Mb

What's the license for users of the server? This wasn't obvious to me when reviewing. Since the IMPUTE2 license restricts commercial use, will commercial users be restricted from using the service? Do the authors anticipate there would be interest from commercial users in using the service?

We found the reviews to be extremely thorough and have addressed suggestions within our revised manuscript. These revisions are additionally described below (in red):

Referee 1:

This manuscript presents the Rice Imputation Server, which is a publically available web application allowing researchers to access a globally diverse rice reference panel to impute genetic information. The claim is that by providing this resource, researchers will be able to capitalize on increased marker density, combine studies more easily, and ultimately identify more precisely polymorphisms relevant for valuable agronomic traits. These claims are well supported from other species such as humans, where imputation has been a standard approach in GWAS for roughly a decade. In concept, the server would be a valuable resource for the rice community; however, there are a number of areas in which additional testing, discussion and guidance would be likely to increase the uptake of this application and hence have greater influence on the field. In particular,

- There is very little discussion or analysis of the effect of having unbalanced distributions of the different subpopulations in the reference panel, nor are the statistics on the makeup of the panel given (though they are probably in prior publications). While the authors mention that this can lead to rare alleles being omitted due to the filtering criteria, it would be good to know whether we expect different imputation efficiencies across the subpopulations. Also, what is the makeup of the gold standard panel of 50 lines in both 3gRK and HDRA? Is the 98% estimate of imputation accuracy based solely on one or two subpopulations (indica/japonica)? This information would be important to understanding the limitations of the imputation.

Many thanks to the reviewer for these valuable suggestions. We agree that additional analyses examining imputation efficiencies across subpopulations are indeed important for rice due to its deep subpopulation structure. Specific changes are described below.

1. We addressed the comment on lack of statistics on the makeup of the panels in the paper by adding a new descriptive figure (**Figure 1**), which displays both the geographical and subpopulation distributions within the 3KRG and the HDRA Panels.
2. The make-up of the Gold Standard Panel was provided in **Supplemental Table 1**. We have added some additional descriptions of the column headers.
3. The reviewer is correct in that the 98% accuracy was based on the Gold Standard panel which is largely composed of *tropical japonica* and *indica* subpopulations. While this accounts for only two subpopulations out of five, *tropical japonica* is the most diverse within the *Japonica* Varietal Group while *indica* is the most diverse subpopulation within the *Indica* Varietal Group. We added additional text in the introduction to provide better background on rice subpopulation for non-rice researchers (lines 65-71). We have also performed subpopulation-specific imputation following guidance from Reviewer #3. These results are found in new Supplemental Figures 4-6.
4. Discussion about subpopulation-specific imputation is found in lines 140-169.

- On a similar note, there is no discussion of the effect of relatedness in the reference panel and/or the data to be imputed. In human studies the reference panels are unrelated, but this is not necessarily true in crops, particularly since the authors do mention the presence of biological replicates in the panels. In particular, regarding the biological replicates – why are they retained? For the gold standard panel, there is hopefully no overlap, but how closely related are they to the rest of the panel (as this may effect the imputation estimates as well)?

The nature of crop species indeed results in panels that often contain related individuals. Presence of biological replicates in the panels are often included in genotyping efforts as technical controls or because there may be many ‘instantiations’ of a single accession that occur as a result of factors such as disparate seed management practices, outcrossing, or re-naming upon adoption across cultures.

Yes, the reviewer is correct in that the samples identical to the Gold Standard Panel were removed prior to using the 3KRG as reference panel to impute the Gold Standard Panel (lines 458-460). Information about relatedness within 3KRG can be found in Supplemental Fig. 3a. As additional assurance, we performed a new round of imputation excluding the most closely related individuals (<10% allele sharing distance), which removed 939 additional individuals from the reference panel, with no change in accuracy. Results are found in Supplemental Fig. 3b-c and cited in lines 137-139.

- In terms of generalizability, the authors mention that the panel can be used to impute GBS-based data, although the example presented is for a SNP array. SNP calling in GBS requires alignment to a reference, and can be more error-prone than SNP arrays. What effect does that have on imputation? Do parameters need to be adjusted? Additional analysis/discussion of this point would be needed before making claims about effectiveness in other data types.

To support our claim, we added an additional analysis found in “The Global *Oryza sativa* Reference Panel (GORP) and Rice Imputation Server” and Supplemental Fig. 12. The analysis was done on individuals genotyped using GBS that were not found in GORP. They were imputed using GORP and validated using an independent re-sequencing dataset. We explain these results in lines 310-320.

- Finally, I am a little concerned about the usage of GORP for imputation since this is already based on one round of imputation (~11K SNPs from 3gRK and 4M+ for HDRA). What effect does this compounding of uncertainty have? Including a comparison of imputation just from 3gRK (which may have lower diversity) vs. imputation from GORP (which has this added level of uncertainty) would help define where potential issues may arise and give more confidence in the approach.

We appreciate this suggestion. We have done so accordingly and the results are summarized in a new figure (Fig. 5) and discussed in lines 309-319. Briefly, we found that GORP improved imputation for markers at low minor allele frequency, which accounts for most SNPs in rice populations.

The statistical approaches and packages used for analysis are appropriate; however, not all parameters are given for the analysis (e.g., ll. 481-482 only states what package was used for analysis – rrBLUP – without providing any information regarding functions or parameters). This will make reproducing the work more difficult. Including R scripts as supplementary files would help to alleviate this issue.

We have added more detail, including functions and parameters, in Methods lines 478-492.

Minor comments:

- Can the framework be used for other crops, given an appropriate reference panel? How modular/customizable is this?

The framework should be extendable to other crop species conditional on reference panel. The study here can serve as a template for the steps taken towards making imputation more accessible to researchers working on other crops, i.e., 1) assemble test datasets; 2) optimize imputation parameters; 3) assess efficiencies and accuracies across subpopulations; 4) evaluate efficiencies and accuracies across genotyping platforms commonly used by the community.

There were several small typos:

- L462 use -> used
OK.

- L165 Average concordance – average or median Previously used median

Correct, this is indeed ‘average’ and not median. These values are direct outputs from IMPUTE2. To reduce potential for confusion, additional clarification is added in lines 172-174.

- Table 2 – are that many digits needed?
Digits have been reduced.

- L75 last sentence does not make sense
The sentence referred to is:

“These panels sample across the geographic and genetic space occupied by *O. sativa*’s five distinct subpopulations.”

We are not sure what part of this was unclear so we kept it. However, we have added additional background regarding the subpopulation of rice in case this helps to clarify our point that the diversity panels are representative of *O. sativa* diversity (lines 65-71).

Referee 2:

In this work, Wang et al. developed a web-based application for imputation analysis in rice GWAS, based on the software IMPUTE2 and the reference panel (e.g., 3000 rice genomes). The application may be useful for association analysis in rice. However, the paper is obviously lack of novelty since the software and the rice reference panel have been published and widely used. Moreover, I don’t think the imputation is a key step in rice GWAS. In fact, the genotyping and follow-up procedures in GWAS (from raw sequence data to high-quality SNP matrix, and from association analysis with mixed model to the identification of candidate genes) include many steps, most of which are more computationally intensive and complicated than the imputation step. Below I have several minor suggestions:

We thank the reviewer for his/her comments, but we politely disagree about the lack of novelty in this study. While it is true that performing GWA can also take many steps, we argue that the increased marker density provided by imputation offers one approach to increase efficiency going from the association analysis phase to the identification of candidate genes. This was demonstrated here using amylose content.

Moreover, the application of imputation extends beyond GWA. It is an approach to help integrate across the numerous disparate genetic datasets and resources that have been (and will continue) accumulating in the rice community. It has the potential to help add value to existing datasets, and to guide additional sampling so that public germplasm panels can better represent global *Oryza sativa* diversity.

1) Why the authors chose IMPUTE2 for imputation? What about other imputation software (e.g., BEAGLE), and what about the situation for heterozygous genotypes?

While there are other software packages available, which other members of our lab group have used, we found IMPUTE2 to be flexible for our needs. BEAGLE4.1 requires all genotypes in the reference panels to be non-missing.

Moreover, IMPUTE4, which is developed to be computationally efficient for haplotype imputation, recently became available. This is an ideal approach for rice genotypes that are nearly 100% homozygous and hence do not require a phasing step. We intend to offer an option to impute with IMPUTE4 as part of the Rice Imputation Server at a future time.

2) Rice has a slow LD decay rate. More examples could be provided in the paper (by the way, *SSIIa* is also a major gene, not minor effect), where the authors need to focus on the analysis of the causative genes/variants (e.g., the prediction of the effects for gene coding) through the integrations of more variation information from imputations.

We apologize but are unsure of the reviewer's meaning here. The overarching goal of our study was not to identify causative genes, but rather to provide a proof-of-concept for using imputation in rice genetics studies and offering a means by which other researchers around the world could easily perform imputation on their own datasets.

3) The result section include many technical details (e.g., the parameter settings of the imputation software), which should be moved into the method section.

The original parameter selection main figure has been moved to the supplemental figures (Supplemental Fig. 2).

4) I did not find the legend for Fig. 5D.

Thank you for finding this. That figure is now Supplemental Fig. 10 and the caption has been added.

Referee 3:

This manuscript describes a public server the authors have built for imputing rice genomes. Additionally, it presents benchmarks of rice imputation accuracy (which help identify appropriate parameter choices for imputation using IMPUTE2), and it also showcases examples demonstrating how imputation can be used to refine rice GWAS associations. The imputation server has a user-friendly interface that is well-documented, so there is a good chance researchers will find it handy.

On the whole, the paper is well-written and does a good job of explaining the utility of the server. I do have several (hopefully constructive) criticisms of the paper that the authors should be able to address as well as some minor suggested improvements.

MAJOR COMMENTS:

1. The authors use concordance between held-out and imputed genotypes to measure imputation accuracy (line 165 and lines 476-477). This metric is generally not very informative because (i) it conflates imputation accuracy across the MAF spectrum and (ii) it awards very high accuracy to rare SNPs under any reasonable imputation algorithm (e.g., always imputing the major allele would achieve genotype concordance close to 1). The authors should instead use the standard approach to benchmarking imputation accuracy, which is to measure R^2 between held-out and imputed genotypes as a function of MAF. (Lower-frequency SNPs are harder to impute and have lower imputation R^2 , so most papers plot a curve of R^2 vs. MAF.)

The reviewer makes very good points and we have exchanged our accuracy metric to r^2 across the MAF spectrum throughout the manuscript per his/her suggestion. These changes are found in:

1. Figure 5
2. Supplemental Figures 2-6 and 12
3. Table 2

2. The authors selected a gold standard set of 50 samples based on overlap between HDRA and 3kRG. This choice is actually problematic because only two subpopulations (*tropical-japonica* and *indica*) are well-represented among the 50 samples. Fortunately, there is really no need to select samples present in both panels. The authors can create a test set from any subset of 3kRG samples by restricting the test set to the 160K filtered HDRA markers, imputing remaining SNPs (using all 3kRG samples not in the test set), and then checking imputation accuracy against the held-out genotypes in 3kRG for the test samples. The authors should perform such an analysis for each of the five subpopulations to determine how well imputation performs on each subpopulation. (Most likely, imputation is harder on AUS.)

We wholeheartedly agree about the importance of assessing subpopulation-specific imputation efficiencies and have done as the reviewer suggested. We found this to be a particularly interesting and worthwhile analysis, and the results are summarized in Supplemental Figures 4-6 and in the text in lines 140-169.

3. Does the combined GORP panel actually add any value for imputation vs. simply using 3kRG? I suspect it doesn't, because the imputed HDRA haplotypes that were added into GORP represent very little additional information: the imputed haplotypes have essentially just been copied over from best matches in 3kRG. The authors should check whether GORP actually provides any improvement in imputation accuracy. They could do so quite easily by comparing imputation accuracy on their 50-sample study set using the GORP reference (minus those 50 samples) vs. the benchmark they already performed using 2973 3kRG samples.

We have addressed this by performing additional imputation with GORP for the IND1 subpopulation, as this was the subpopulation that performed the least favorably with the 3KRG. Results are summarized in Figure 5 and described in lines 288–308.

4. I'm not sure what to make of the haplotype analysis starting at line 217. Generally, haplotype analysis of greatest interest when the true causal variant is untyped and unimputed -- but here, there is a clear causal variant in the region (the *intron1* splice variant) and as far as I can tell, that variant may explain the entire association signal. A conditional analysis is really needed here: the authors note that 22 SNPs in the *Wx* genic region significantly associated with amylose content, but before embarking on haplotype analysis, how many of those associations are simply explained by LD to the *ms*SNP? To get at this question, the authors can simply condition on the lead SNP and see how many associations disappear (with the caveat that the lead SNP is likely to be imperfectly imputed, so remaining associations still need to be interpreted with care). My guess is that many of the 22 associations will turn out to be explained by LD. For example, the authors claim that the second FNP at 1,768,998 is "associated with amylose content" (line 251), but based on $r^2 \sim 0.18$ (Supp Fig 7) to the lead FNP which has $-\log_{10}p = 64.8$, I would expect a $-\log_{10}p$ of $0.18 * 64.8 = 11.7$, which looks to be exactly what is observed in Figure 4 at the second FNP -- suggesting that in fact this signal is just LD.

The reviewer makes good points about the additional associated markers being a function of LD. We had originally intended the gene haplotype analysis to be a demonstration of what a user could potentially do with the imputed HDRA dataset versus the original HDRA dataset that did not include as many genic SNPs in *Wx*. This analysis would be most useful, as the reviewer states, in scenarios when the FNP is untyped and unimputed. In light of the reviewer's suggestions that haplotype analysis does not add much additional information in this case, but still wanting to include evidence showing that this type of analysis is possible using the imputed dataset, we made the following changes:

1. Performed the association analysis conditioned on the FNP as suggested (summarized in Results lines 255-260; Fig 4c)
2. Moved the original main figure displayed the haplotype analysis to supplemental figure 10.
3. Trimmed the text in the results section (from 52 lines total to 27 lines total).

MINOR COMMENTS:

Table 1: "Haplotype number" is confusing because the genomes are diploid. How about "Unique samples" instead?

Sure, we have made this change in the table.

Table 1: Why does GORP have a larger number of markers than 3kRG?

The GORP (previous GORP from original submission) had a larger number of markers than 3kRG because out of the ~160K SNPs used on the HDRA study panel, ~11K were not found in the 4.8M 3kRG dataset.

The new GORP, generated using the 4.8M 3kRG and full 700K HDRA dataset, is at 5.2M markers, again greater than 3kRG, due to the markers found in HDRA not present in 4.8M 3kRG. The panel was assembled via reciprocal imputation in IMPUTE2, so markers absent in each dataset are imputed, resulting in a greater marker number than the larger of the two panels.

Line 117: "O. sativa is an inbreeding species such that genotypes are generally already equivalent to haplotypes" -- quantification of this statement would be helpful (e.g., reporting median and range of per-sample heterozygosity, or better yet, a box plot).

Median homozygosity in the HDRA Panel (700K SNPs) and the 3kRG Panel (4.8M SNPs) are 98.6% and 99.5%, respectively. We have made boxplots to display the range as suggested and added them to the new Figure 1.

Line 133: I was surprised that imputation accuracy appeared to max out at $R=1500$. In human data sets, accuracy always steadily increases with reference panel size. Does this result still hold after measuring accuracy using R^2 and stratifying by MAF? If so, can the authors suggest why?

After switching to the r^2 accuracy metric, the results still hold (Supplemental Fig. 2d). We suggest that the differentiation between the results shown here for rice and the results observed for human genetic imputation is fundamentally linked to the idea that there are groups of germplasm that are closely related to each other. Unlike human populations, rice is a naturally inbreeding species with extended LD, and as a product of both natural and artificial selection, there is a high degree of haplotype-sharing within subpopulations (Supplemental Fig. 3a). This is particularly true for accessions from the same geographic/ecological zone where both fitness and the forces of human selection tend to fix certain haplotypes.

Line 165: It's unclear to me what samples were used to measure "average concordance": the 50 study panel samples? (Again, MAF-stratified R^2 would be much more informative here.)

We apologize that this was unclear. "Average concordance" is directly computed by IMPUTE2 and outputted in the ".summary" files. As described by IMPUTE2's output, this analysis:

“is based on an internal cross-validation that is performed during each IMPUTE2 run. For this analysis, the program masks the genotypes of one variant at a time in the study data (Panel 2) and imputes the masked genotypes by using the remaining study and reference data. The imputed genotypes are then compared with the original genotypes to produce the concordance statistics shown in the table. You can learn more about this procedure and the contents of the table at http://mathgen.stats.ox.ac.uk/impute/concordance_table_description.html.”

For our own computations, we have now replaced with r^2 stratified across MAF as suggested by the reviewers (see reply to this reviewer’s first major comment).

Lines 200-201: "LD, not the number of markers, likely limits the resolution of GWAS in rice" and 317-319: "extensive linkage disequilibrium is the primary constraint to narrowing down association regions rather than the number of markers per se." I think these statements are unnecessarily negative; they seem to argue against the utility of the imputation server! Presumably the authors are referring to the fact that the width of the association peaks remain the same after imputation (as they must). However, in the best case scenario -- which the authors demonstrate here -- imputation can still help fine-map association regions when the true causal variant is successfully imputed and becomes the most significant association (as the authors show at Wx and SSIIa; Figure 3 is very nice).

We appreciate the reviewer’s kind feedback. We like Figure 3 as well – credit for the base code goes to the Diabetes Genetics Initiative of the Broad Institute.

While we certainly agree with the reviewer’s comments here, we did end up keeping one of the two sentences referred to because we want to make sure that our audience also understands this as there appears to be some mystique surrounding what one can and cannot do with GWAS among the rice community.

Figure 4: Adding Supplementary Figure 7 (LD and MAF) as a bottom panel of this figure would greatly help its interpretation.

We agree and have added this panel to Figure 4.

Line 255: Supp Fig 7 should be referenced here instead of Supp Fig 6.

Thank you for noticing this. We have now referenced the correct figure (currently modified in lines 255-256 and displayed in [main] Figure 4d).

Figure 5: Caption for panel D is missing.

We have corrected this mistake (currently Supplemental Fig. 10d).

Line 282: What's the LD between the two adjacent SNPs?

The LD (r^2) is 1.0. Out of the samples examined for GWAS, there was only 1 recombinant.

Discussion: I would recommend shortening the Discussion.

The discussion is significantly shorter now at 65 lines versus the previous 114 lines. The manuscript now complies with the 5,000 word limit [Intro+Results+Discussion].

Line 487-488: Why were no PCs necessary for the subpopulation-specific analyses? The top PCs appear

to be explaining large amounts of genetic variance (>5%), assuming that's what the y-axes of Supp Fig 6 are measuring. (The meaning of the y-axes should be clarified in Supp Fig 6.)

We followed the approach used in McCouch et al. 2016 who developed the original HDRA (700K SNP) dataset and GWAS strategy for rice. Based on experience in her lab, the use of 1 PC in the subpopulation-specific GWAS did not affect the results of the analysis compared to when 0 PCs were used. Therefore, the recommendation was to use 0 PCs for these analyses as the most straightforward approach.

The meaning of the y-axes are now clarified in the figure legend (currently Supplemental Fig. 13).

Referee 4:

In this work, the authors present the Rice Imputation Server, a publically available web server that provides imputation resources to the rice genetics community. The goal of the paper is to facilitate genetics research in rice by providing a resource for imputation of rice accessions using a globally diverse reference panel. They describe a combined panel made of two sub-reference panels, one the HDRA panel from a 770K array using approx 160K filtered high quality SNPs and approx 1500 samples, and the 3kRG panel from 29M to 4.8M filtered high quality SNPs and approx 3000 samples. The paper is extremely well written and well presented and most analyses are appropriately thorough. The benefit for downstream interpretation of imputation is particularly well presented (e.g. Fig 3).

My two biggest worries about this paper are first, is it really true that high computational demand is limiting uptake of imputation in rice, necessitating this server as stated in the abstract (vs a download link for the reference panels), and second, is imputation from arrays or GBS the best long term strategy for genetics research in rice (and hence, how long will this server truly benefit the community)? On the first point, my understanding with human data is that human phasing and imputation servers arose at least partially out of an inability to release the full HRC reference panel data due to privacy / consent reasons, so imputation servers are the only way for human genetics researchers to obtain the full utility of the complete HRC panel. I assume no such restriction holds here, and the reference panel sizes proposed here are comparable to those used in human genetics research from ~2009-2015 that were imputed by hundreds (thousands?) of groups worldwide, and would not strike me as beyond the capability of a normal lab that sought to perform GWAS. On the second point, either a) you're interested in prediction, so you don't care what the causal SNP is, so you can use a low density marker array and don't really need imputation, or b) you're interested in the biology. For b), you really want high sensitivity to detect variants, to make sure you've captured the causal variant (FNP). The rice genome is comparatively small (430 Mbp), which makes resequencing an attractive option for genotyping. If I were a researcher today and wanted to perform GWAS and fine-mapping in 1,000 rice samples from a single population tomorrow, would I choose arrays and imputation? If making libraries can be done for <20USD, and if we believe 100 Gbp / 1000 USD (human 30X for 1000 USD), then 1 Gbp / 10 USD, so you can get a little over 2X low coverage for 30 USD (so maybe 60 dollars at scale in a commercial lab) and you can use Beagle / STITCH / etc for imputation without a reference panel and get high quality genotypes for all SNPs. Are arrays (at the density cited here of 700K?) and reference panels cost and time-competitive and the way forward?

We thank the reviewer for this thoughtful comment. We also have spent time thinking about the potential utility that an imputation server would offer to our research community. One of the points we emphasize in this revised submission is that there are several different avenues of application: the first is improvement to GWAS, the second is ability to integrate disparate genetic datasets, and the third is to help gene bank managers develop a rational strategy for genomic characterization of the tens of thousands of accessions in their collections.

The first application involves labs doing GWAS, and most of them certainly would be able to undertake imputation internally without an imputation server. For that application, we initially thought about simply releasing the GORP in IMPUTE2 reference panel format along with imputation parameters, without building an application. It would have certainly been easier! But the GORP is 90GB when formatted in the IMPUTE2 phased reference panel format, and the size alone would have prohibited most smaller groups from being able to use imputation. There exists wide variation in the rice genetics and breeding community with regards to institutional (and particularly, computational) resources. Being part of more well-endowed research groups (at Cornell University and the International Rice Research Institute), it is part of our responsibility to generate tools that can help bridge the gap between well-resourced and under-resourced labs and research communities.

The second application is arguably more essential for the rice community and for many other plant communities as well. Imputation allows the immediate integration of datasets where different types of germplasm and collections of samples were used to generate genomic datasets of varying marker densities. Imputation facilitates the integration of these resources by mapping them onto a common genomic framework without the added time and expense of re-genotyping. We added text in the introduction (45-57) and the discussion (386-399) to try and flesh out this point.

The third application revolves around gene bank managers working with plant genetic resources at the local, national and international levels. Gene banks are notoriously underfunded and poorly resourced, particularly local and national genebanks in the developing world where most plant genetic resources are found. The rice imputation server is likely to be of interest to gene bank managers who seek to develop rational strategies for characterizing their collections using scarce resources. As described in the discussion (368-385), it can provide a valuable framework for prioritizing which accessions to sequence first, and at what depth of coverage, to maximize benefit and minimize cost.

I also have several questions and comments about the paper as presently written.

Major comments:

1) In Results section "Parameter selection for rice imputation", you are selecting parameters by using HDRA samples with imputation from a 3kRG reference panel. But the whole point of the paper is providing an imputation server with a reference panel, and that panel uses HDRA and 3kRG together. Obviously HDRA + 3kRG shouldn't be worse than 3kRG, but is it any better? To verify this, I would be interested in knowing a) does adding HDRA to 3kRG improve imputation, and b) does this result hold if you remove members from that sub-population from within HDRA? Specifically, I would like to see, for each of IND, AUS, TEJ and TRJ the imputation of the members of the overlap from that subpopulation, using a combined reference panel of (3kRG minus overlap) + (HDRA minus overlap minus subpopulation), as compared to (3kRG minus overlap).

This might be too exhausting for all the parameter benchmarking, but I would do this with final parameters somewhere around where you present Table 2.

We have now compared the performance of imputation using GORP versus using 3kRG alone, following the scheme that the reviewer suggested for IND1 subpopulation as it had the most room for improvement. Please see new Figure 5 and lines 289-308.

Also, just to be clear, other than identical samples, are there any other relatives between the overlap set and either HDRA and 3kRG, with long tracts of shared IBD that would artificially inflate imputation accuracy estimation?

We thank the reviewer for this point and have now integrated an additional analysis for this (Supplemental Figure 3). We saw high imputation accuracy even after removing nearly 1000 of the most closely related lines to the Gold Standard Panel.

2) *The discussion ~340-374 worries me. First, can the authors (somewhere in the paper) give a slightly more detailed breakdown on how 700K -> 160K in the context of the 29M / 19M / 4.8M SNPs from the 3kRG? How many of the array SNPs are removed for frequency, quality or intersection reasons? Also, I don't really understand why the requirement to have >1% frequency in the GORP exists, given you've shown this to be a problem in the Discussion? Why not fix this now instead of later by considering >4.8M SNPs (at least extend 4.8M to include >1% MAF any sub-population to prevent this specific problem)? Finally, this makes me question how time saving an imputation server will be, if follow up work requires downloading all the reference panels to your computer anyway to look at subpopulation specific haplotypes, how much effort are you really saving by performing a first imputation in the cloud.*

We have now added a more detailed overview of the datasets used in this study in Supplemental Figure 1.

We alleviated part of the MAF issue in this revision. In the current assembly of GORP, we used the 700K HDRA Panel without filtering (lines 325-326; 474-478), however, the high-quality 4.8M SNP dataset on the 3kRG Panel is only available as a “pre-filtered” dataset so we are unable to address the 0.01 global MAF issue. This remaining caveat is still pointed out in Discussion lines 359-367.

Follow-up work does not necessitate downloading of the entire GORP. Regional data (such as at a locus of interest resulting from GWAS) can be extracted and downloaded using SNPSeek (<http://snpseek.irri.org/>). Additionally, it is the GORP phased Reference Panel format where size is likely to be a barrier (90GB) to its utility, whereas the plink binary format comes in at a much more reasonable 12GB. Users of the rice imputation server can interact and do analyses with the entire GORP simply by downloading the plink binary data. Thus, having access to the GORP phased Reference Panel via the imputation server allows them to avoid dealing directly with the 90GB reference panel; that would only be necessary if they wanted to perform imputation using GORP on their own machines.

We provide the plink binary GORP fileset at RiceDiversity.org and European Variants Archive. SNPSeek can be used to extract region-specific information on GORP.

These data repositories are now specified in 507-511. Note that we are awaiting an accession number from European Variants Archive.

3) *Can you please switch accuracy to per-site squared correlation (methods 475-477, Figure 1, etc). Correlations are much more informative for interpreting power, and it can be hard to interpret accuracy without precise knowledge of the allele frequency spectrum (e.g. if you imputed a dataset with only 1% MAF SNPs completely at random your accuracy would be approx 98% but your imputed dataset would be completely useless)*

Yes, thank you for the suggestion. Please refer to the response to Reviewer #3, major comment #1.

4) *It seems that a previous imputation server (the Michigan human one) also provides an open source framework for building imputation servers*
<https://www.nature.com/articles/ng.3656>
<https://imputationserver.sph.umich.edu/start.html> (this website has a part saying “Build your own imputation server”). Even if this doesn't end up in the paper, I'd be curious to hear a little more about why the authors chose to go with their own server vs build off someone else's work

That's a good question. To be honest, we had already explored parameter settings for rice imputation using IMPUTE2 and were very pleased with its results in rice, and had already begun efforts building the Rice Imputation Server before we found the Michigan Imputation Server (which looks to be very user-friendly for the human genetics community). From what we understand, the Michigan Imputation Server utilizes its own imputation engine (with the algorithm improvements described in its paper), while we were already very familiar with using IMPUTE2. And so, instead of going through parameterization efforts again with a new imputation pipeline that we were unfamiliar with and tossing the work we had already done with the application, we chose to continue developing RIS. We are pleased with the outcome, especially with the updates made according to this revision, and hope that its interface is intuitive for users in our community.

Minor comments:

Reading Table 1 and lines 300-302, it would seem that 3kRG was imputed into HRDA to create a haplotype reference panel with 4.829M SNPs with 4591 samples (4481 haplotypes). But reading the Methods 477-480 and the IMPUTE2 website, it seems the "merge reference panel" option is used which keeps the two panels separate. Which one does the server actually use? On the server is a single reference panel used or two of them? If a single reference panel, can the authors explain somewhere in a sentence why (does this make subsequent imputation faster? More accuracy?) and specifically how they they do this (i.e. X HDRA SNPs into Y 3kRG SNPs with parameters W, Z, etc)

Sorry about this confusion. The reviewer's deductions are partially correct. Yes, GORP was assembled using the merge reference panel option of IMPUTE2. No, this does not mean that the reference panels are kept separate. They are outputted as a single panel. From the IMPUTE2 website:

*"If you simply want to merge two reference panels without imputing missing genotypes in a study dataset, you should add the **-merge_ref_panels_output_ref** or **-merge_ref_panels_output_gen** flag and omit the study genotypes (**-g** or **-known_haps_g** file) from your IMPUTE2 command"*

The Rice Imputation Server uses this single panel (in a phased format to make imputation much faster than using unphased reference panel). While IMPUTE2 does have options to use multiple panels that are kept separate, it is faster to use a single phased reference panel because the first two steps in imputation via two separate reference panels is to reciprocally impute each panel. These first two steps are skipped if we use a single phased reference panel directly. We have added text to the Methods in lines 474-478 that hopefully clarifies this.

Table 2 could probably go to the supplement.

We agree. It is now Supplementary Table 2.

My imputation job REC_V54VZP of the test data failed with Status: [u'FAILURE', u'PENDING', u'SUCCESS']. Better error reporting is needed (and maybe remove the "u"'s from your JSON / YAML), and obviously a paper on Rice Imputation Server needs to be able to actually impute rice data! I also uploaded a PDF (Status REC_V55VP8 [u'PROGRESS', u'PENDING', u'SUCCESS']) and during the several hours I spent reviewing this paper, the job never failed. It would be extremely useful for users if you could validate all your inputs quickly upon submission (is it a tarball? Does it contain the right number of files? Are the file extensions correct? Do the files conform to input requirements?), as I think anyone who has spent time in bioinformatics will know how much time is wasted debugging discrepancies between the input data that has been prepared and what the program accepts. Centralization of this process would actually be a major plus of this server (centralized debugging).

We are grateful to the reviewer for catching this and have now improved input validation. To validate user input files, we have made changes to the backend that enforce the current input requirements via several checkpoints that

1. Checks for files with expected naming structure
2. Checks to ensure that there are 12 chromosomes (12 .gen and 12 .sample files) present with expected user-defined base name
3. Inspect .gen files to ensure that chromosome number listed matches expected chromosome number in file name
4. Outputs more informative error message if input validation fails. These error messages indicate which of the input requirements are not met by the user-provided input file.

To make imputation progress more transparent to the user, we have consolidated the progress output and replaced it with a progress bar. This progress bar activates after the input files have been validated and the imputation pipeline has begun.

Finally, we have made modifications within the RIS tutorial that clarifies input file preparation:

1. Additional section on how to tar fileset (tutorial)
2. Clarification on fileset naming scheme (tutorial)

Is the recombination rate in Fig 3 as expected? From 1.8 to 2.0 Mbp, recombination rate is >4 cM/Mb? This seems implausible in the context of the discussion talking about high LD in these mapping populations, and Fig 3C which suggests recomb hotspots at ~1.55, 1.62, 1.8 and 1.98 Mbp. I would love it if my mapping species all had uniform recombination rates of >4 cM/Mb

Sure, one factor that affects LD is recombination rate. However, the high LD we observe in rice is also a function of its inbreeding nature (such that physical recombination events go undetected, see Figure 1 for degree of per-sample homozygosity) and selection. Here, we derived our recombination map using genomic data on a bi-parental Recombinant Inbred Line population derived from a cross between *Indica* and *Japonica*. Average genome-wide recombination rate in *Oryza sativa* varies depending on the report but is generally around 3.5 to 5.5 cM/Mb. And of course, this rate varies across the genome.

References:

1. IR64 x Azucena RIL population recombination rates in Spindel et al 2013: Bridging the genotyping gap: using genotyping by sequencing (GBS) to add high-density SNP markers and new value to traditional bi-parental mapping and breeding populations
2. IR64 x Nipponbare RIL population recombination rates in Jander 2015: The Tyrosine Aminomutase TAM1 Is Required for b-Tyrosine Biosynthesis in Rice
3. Si et al 2015: Widely distributed hot and cold spots in meiotic recombination as shown by the sequencing of rice F2 plants
4. Chen et al. 2002: An integrated physical and genetic map of the rice genome
5. Kurata et al. 2002: Rice Genome Organization: the Centromere and Genome Interactions

What's the license for users of the server? This wasn't obvious to me when reviewing. Since the IMPUTE2 license restricts commercial use, will commercial users be restricted from using the service? Do the authors anticipate there would be interest from commercial users in using the service?

To address the licensing issue, we followed the same protocol as is used by the human genetics community when using IMPUTE2. On the RIS website, there is a link called "Terms of Use" which takes users to a page outlining our terms and conditions (copied here):

The Rice Imputation Server (RIS) is licensed only for non-commercial and academic use. RIS utilizes IMPUTE2 as the imputation engine; only academic and non-commercial users that are eligible to use IMPUTE2 can make use of RIS. IMPUTE2 is freely available for academic use. For non-academic purposes, refer to the IMPUTE2 license.

The terms and conditions contains direct links to the IMPUTE2 license and IMPUTE2 website.

In general we found the reviewers' feedback to be very valuable, especially in directing deeper analyses regarding sub-population specific imputation and offering a more appropriate metric for assessing imputation accuracy. We have made substantial revisions to the manuscript according to these suggestions and believe the outcome is an improved study from our previous submission.

Reviewers' comments:

Reviewer #1 (Remarks to the Author):

I'm satisfied with the authors' response to reviews.

Reviewer #3 (Remarks to the Author):

The authors have done an excellent job with their revision, which is comprehensive and substantially improves an already high-quality manuscript.

I have a just one minor suggestion that the authors may consider at their discretion. The authors have addressed my previous comment 3 (about comparing GORP vs. 3kRG imputation performance) with suitable analyses, but their interpretation of the results (lines 302-309) seems to more strongly favor GORP than the data warrants. They might consider rewording these sentences. A separate minor point is that the figure references in these sentences need to be updated; Fig. 5 should now be referenced instead of Fig. 6, and it doesn't have a panel (c).

Reviewer #4 (Remarks to the Author):

The authors have done a very thorough job with their responses and subsequent modifications to the manuscript which is commendable. I am largely satisfied but have a few follow up points.

Round 2 major comment 1

Methods lines 473 to 475 state "Accuracy (marker-based r^2 between imputed and true calls) was assessed using custom R scripts and calculated as the number of exact matches between imputed and original (i.e. re-sequenced) samples divided by the number of markers typed in the original sample.". I'm now confused, which is being done here? Are you calculating r^2 (the squared Pearson correlation between imputed genotypes and truth genotypes (not using missing truth genotypes)) (also imputed dosages would seem more sensible than imputed genotypes) or the measure of accuracy which is really concordance you are describing "the number of exact matches between imputed and original (i.e. re-sequenced) samples divided by the number of markers typed in the original sample."? I would strongly suggest removing accuracy for the paper entirely in favour of correlation if not already done so. I assume some figures / results still use the accuracy / concordance, as this would make Supp Figure 2A make a lot more sense (see below).

Round 2 minor comment 1

Supp Figure 2A phased vs unphased looks like they're identically the same figure? Please double check to be 100% sure this isn't a bug. Also, I am very confused about why there seem to be 3 lines: one at 1, one trending towards ~ 0.97 on the right, and one trending towards ~ 0.9 on the right. This is per-site correlation between imputed dosage (values [0-

2]) and truth genotype ($\{0, 1, 2\}$), right? For B, this sentence appears to be repeated "For all parameter combinations, median accuracy (computed as site squared correlation) were 1.0." Anyway I would suggest just giving the mean and not the quantiles.

Follow up to Round 1 major comment 1

Lines ~300-308 of the Results references Figure 6 but I think this should be Figure 5. For Figure 5, I would suggest replacing the boxplots and just plot the average, and combine A and B into one panel. I'm not a great fan of the smoothing here because there's no obvious theoretical distribution that this should follow. I assume the r^2 is site-wise based on its use elsewhere in the text but if not please note. How many individuals were used for the testing here? I would make the MAF plot bigger as a reader, it's difficult to infer things based on this (in it's context in the text, I was curious to know what % of SNPs had $MAF < 0.158$, which I would like to have been able to infer from reading the plot. Looks like 90%?). Finally overall it would have been nice if the extra reference panel led to a more substantial gain in accuracy, from this Figure GORP seems to add only ~2-5 percentage points in performance for $MAF < 0.10$ SNPs.

Follow up to Round 1 minor comment 3 (on imputation on the server)

I tried the server again. On my first attempt "f7fff323-faf3-4eeb-94c8-f372a1632e72", I got an error and no explanation. The tarball name had a space in it, perhaps that was the problem? I also never got an email that my job crashed (why do I need to enter an email if I'm not going to get email feedback?). My second job REC_5W2P1PV was stuck at 1.59% for the length of time it took me to write this re-review. I eventually aborted when I finished writing this review. Is one supposed to be able to log out / log back in? When I launched a new website in a new browser window I couldn't find a record of my "running" job. Anyway, these are all problems that are solvable over time. I'm glad to read in your response you implemented checking of input file formats etc. It's probably a good idea to beta-test this with some external sites and/or friends to find any more residual bugs and make sure the server is working smoothly.

Thank you again for reviewing our manuscript entitled “Rice Imputation Server: a platform to enable efficient integration of rice genetic resources”. We appreciate the rapid response and have addressed the additional suggestions made by Referees #3 and #4 within our revised manuscript. A point-by-point response may be found below (in red):

Referee 3:

The authors have done an excellent job with their revision, which is comprehensive and substantially improves an already high-quality manuscript.

I have a just one minor suggestion that the authors may consider at their discretion. The authors have addressed my previous comment 3 (about comparing GORP vs. 3kRG imputation performance) with suitable analyses, but their interpretation of the results (lines 302-309) seems to more strongly favor GORP than the data warrants. They might consider rewording these sentences. A separate minor point is that the figure references in these sentences need to be updated; Fig. 5 should now be referenced instead of Fig. 6, and it doesn't have a panel (c).

Many thanks to the reviewer for these very kind comments. We have taken his/her suggestion into consideration and have modified the relevant lines accordingly (currently lines 328-335).

We have fixed the incorrect figure references. Figure 5 has a panel c (embedded in panel b so perhaps this was overlooked).

Referee 4:

The authors have done a very thorough job with their responses and subsequent modifications to the manuscript which is commendable. I am largely satisfied but have a few follow up points.

Round 2 major comment 1

Methods lines 473 to 475 state “Accuracy (marker-based r^2 between imputed and true calls) was assessed using custom R scripts and calculated as the number of exact matches between imputed and original (i.e. re-sequenced) samples divided by the number of markers typed in the original sample.”. I'm now confused, which is being done here? Are you calculating r^2 (the squared Pearson correlation between imputed genotypes and truth genotypes (not using missing truth genotypes)) (also imputed dosages would seem more sensible than imputed genotypes) or the measure of accuracy which is really concordance you are describing “the number of exact matches between imputed and original (i.e. re-sequenced) samples divided by the number of markers typed in the original sample.”? I would strongly suggest removing accuracy for the paper entirely in favour of correlation if not already done so. I assume some figures / results still use the accuracy / concordance, as this would make Supp Figure 2A make a lot more sense (see below).

We thank the referee for once again providing a comprehensive review of our work.

We do apologize for any confusion. In our previous revision, we indeed changed all accuracy analyses throughout the manuscript to per-site squared correlation as suggested by multiple reviewers, such that there were no remaining analyses that still used the original genotype-based accuracy metric. As for ‘concordance’, this is stated explicitly and referred to exclusively as ‘concordance’ per IMPUTE2 output naming, and not ‘accuracy’. However, we apparently failed to remove the latter part of the sentence in the Methods to which the reviewer refers; this was an accidental remnant during the revision process. We have eliminated this now and thank the reviewer for catching this (lines 526-528).

Please see response below to Round 2 minor comment 1 regarding Supp. Figure 2A.

Round 2 minor comment 1

Supp Figure 2A phased vs unphased looks like they're identically the same figure? Please double check to be 100% sure this isn't a bug. Also, I am very confused about why there seem to be 3 lines: one at 1, one trending towards ~0.97 on the right, and one trending towards ~0.9 on the right. This is per-site correlation between imputed dosage (values [0-2]) and truth genotype ({0, 1, 2}), right? For B, this sentence appears to be repeated "For all parameter combinations, median accuracy (computed as site squared correlation) were 1.0." Anyway I would suggest just giving the mean and not the quantiles.

Regarding Supp. Figure 2A, thanks to the reviewer for this comment. We double-checked and found that there was a bug: our original script pulled out unphased data for both phased and unphased plotting. We have fixed this and this panel is now updated with the correct figure. We apologize about the mistake and are grateful to the reviewer for catching this.

Yes, we did use per-site squared correlation between imputed genotypes and truth genotypes, but the format for imputed genotypes was not in dosage format (values [0-2]) but rather genotype format ({0, 1, 2}), with the rationale that this is the format that would be used for GWAS and other downstream analyses by users. This was done by converting IMPUTE2 imputed dosage genotype files into plink format. This additional clarification has been added to Methods lines 526-528. We are not entirely sure why accuracy should show multiple trends in relationship to increasing MAF, but in light of the reviewer's remark on dosage, we speculate whether this could be related to using genotypes rather than dosage.

We have removed the redundant sentence mentioned.

Follow up to Round 1 major comment 1

Lines ~300-308 of the Results references Figure 6 but I think this should be Figure 5. For Figure 5, I would suggest replacing the boxplots and just plot the average, and combine A and B into one panel. I'm not a great fan of the smoothing here because there's no obvious theoretical distribution that this should follow. I assume the r^2 is site-wise based on its use elsewhere in the text but if not please note. How many individuals were used for the testing here? I would make the MAF plot bigger as a reader, it's difficult to infer things based on this (in its context in the text, I was curious to know what % of SNPs had MAF <0.158, which I would like to have been able to infer from reading the plot. Looks like 90%?). Finally overall it would have been nice if the extra reference panel led to a more substantial gain in accuracy, from this Figure GORP seems to add only ~2-5 percentage points in performance for MAF <0.10 SNPs.

We thank the reviewer for his/her comments. Figure 5 with double panels as presented was indeed the result of consideration of the very points the reviewer has raised. We outline our rationale below:

We agree that the relationship between r^2 and MAF does not follow an obvious function, only that it should theoretically improve with increased MAF and likely non-linearly. The rationale behind 5a is for presenting the actual data while the reasoning for 5b is for ease of visual comparison versus 5a. While we certainly would like to consider the reviewer's suggestions for this figure for plotting means, it is not obvious how to do so. In order to get mean r^2 across the MAF spectrum, we would need to bin the data into MAF bins first (such as was done for Figure 5a). Yet, means are not very good representatives of the distribution within each bin. This was the original reason for presenting the boxplots in Figure 5a—they capture the distributions within-bin and do not assume any distribution (as the boxplots reflect quantiles). Additionally, averages may be sensitive to outliers whereas medians are not. For these reasons, we would like to retain our original figure that shows both the distribution of original data as well as fitted curves.

We have now indicated the proportion of SNPs with $MAF < 0.158$ in lines 330-331. A larger version of Figure 5c is found in Supp. Figure 5b, and we have now cross-referenced this within the figure legend for 5c so that interested readers may be able to take a closer inspection.

Regarding the comment on GORP: Referee #3 made a similar comment, therefore we have adjusted our sentences accordingly in lines 328-335.

Follow up to Round 1 minor comment 3 (on imputation on the server)

I tried the server again. On my first attempt “f7fff323-faf3-4eeb-94c8-f372a1632e72”, I got an error and no explanation. The tarball name had a space in it, perhaps that was the problem? I also never got an email that my job crashed (why do I need to enter an email if I’m not going to get email feedback?). My second job REC_5W2P1PV was stuck at 1.59% for the length of time it took me to write this re-review. I eventually aborted when I finished writing this review. Is one supposed to be able to log out / log back in? When I launched a new website in a new browser window I couldn’t find a record of my “running” job. Anyway, these are all problems that are solvable over time. I’m glad to read in your response you implemented checking of input file formats etc. It’s probably a good idea to beta-test this with some external sites and/or friends to find any more residual bugs and make sure the server is working smoothly.

Thanks again for taking the time to beta-test our application. Upon review of our logs, we have added these additional changes to ensure proper file set input in the event users change their tar file names after compression.

- 1) The app now works if the tar file name and the uncompressed directory names are different. This was causing an issue before that the reviewer picked up.
- 2) We have incorporated fixes so that the app now handles tar file names that include spaces or other symbols, although we decided to still retain the original directions that inform the user not to include spaces in the file name.
- 3) We added feedback so that when an error occurs, the user will be directed to a webpage that displays the error message; the user can then revise their input and retry, or notify us that something has happened and show us the error message.

In response to the reviewer’s comment regarding the email feature for this application: We specifically turned off the email functionality for the review process to ensure that the identities of reviewers were protected in the case that any referee mistakenly entered his/her email address. This feature will be turned on once the review process is done. The user will receive emails for the following steps:

- 1) Indicating that his/her fileset has been uploaded and validated successfully and a locator ID with link to follow the status of their job. Alternatively, an email indicating that the upload was not able to be validated.
- 2) Indicating that the imputation has started processing the fileset. This may be immediately following the first email or at a later time depending on the job queue.
- 3) Indicating that the imputation pipeline is completed with a download link.

In addition to the above revisions, we made two minor decisions to rename the ‘GORP’ to ‘RICE-RP’ (Rice Reference Panel) and update the acronym ‘3kRG’ (3,000 Rice Genomes) to ‘3KRG’ throughout the manuscript, based on suggestions by co-authors.

We are especially grateful to Referee #4 for his/her thorough review and helping us to make these latest corrections. We believe that these latest revisions have improved our work and look forward to hearing from you.